# RPPUF: An Ultra-Lightweight Reconfigurable Pico-Physically Unclonable Function for Resource-Constrained IoT Devices

**Zhao Huang** [1], **Liang Li** [1], **Yin Chen** [2], **Zeyu Li** [1], **Quan Wang** [1,*] **and Xiaohong Jiang** [3]

1 School of Computer Science and Technology, Xidian University, Xi'an 710071, China; z_huang@xidian.edu.cn (Z.H.); lliang@stu.xidian.edu.cn (L.L.); zeyuli@stu.xidian.edu.cn (Z.L.)
2 Graduate School of Media and Governance, Keio University, 5322 Endo, Fujisawa 252-0882, Japan; yinchen@sfc.keio.ac.jp
3 School of Systems Information Science, Future University Hakodate, Hakodate 041-8655, Japan; jiang@fun.ac.jp
* Correspondence: qwang@xidian.edu.cn; Tel.: +86-1399-132-7112

**Abstract:** With the advancement of the Internet of Things (IoTs) technology, security issues have received an increasing amount of attention. Since IoT devices are typically resource-limited, conventional security solutions, such as classical cryptography, are no longer applicable. A physically unclonable function (PUF) is a hardware-based, low-cost alternative solution to provide security for IoT devices. It utilizes the inherent nature of hardware to generate a random and unpredictable fingerprint to uniquely identify an IoT device. However, despite existing PUFs having exhibited a good performance, they are not suitable for effective application on resource-constrained IoT devices due to the limited number of challenge-response pairs (CRPs) generated per unit area and the large hardware resources overhead. To solve these problems, this article presents an ultra-lightweight reconfigurable PUF solution, which is named RPPUF. Our method is built on pico-PUF (PPUF). By incorporating *configurable logics*, one single RPPUF can be instantiated into multiple samples through configurable information *K*. We implement and verify our design on the Xilinx Spartan-6 field programmable gate array (FPGA) microboards. The experimental results demonstrate that, compared to previous work, our method increases the uniqueness, reliability and uniformity by up to 4.13%, 16.98% and 10.5%, respectively, while dramatically reducing the hardware resource overhead by 98.16% when a 128-bit PUF response is generated. Moreover, the *bit per cost* (BPC) metric of our proposed RPPUF increased by up to 28.5 and 53.37 times than that of PPUF and the improved butterfly PUF, respectively. This confirms that the proposed RPPUF is ultra-lightweight with a good performance, making it more appropriate and efficient to apply in FPGA-based IoT devices with constrained resources.

**Keywords:** pico-PUFs; ultra-lightweight; reconfigurable; hardware security; FPGAs

## 1. Introduction

With the advancement of the Internet of Things (IoT) technology, billions of devices are connected to the internet to provide services in our daily lives. It can be said that the IoT is gradually becoming an important part of the new infrastructure. However, the explosion of IoT devices in various industries has raised concerns about security issues [1–3]. This is primarily due to the fact that IoT devices are usually deployed in unattended open places and are connected to each other through wired or wireless networks, providing adversaries more opportunities to tamper with them maliciously [4–6]. Figure 1 presents the security threats faced by each layer of the IoT framework [7,8]. From the perspective of the IoT framework, it can be divided into four layers: a device/physical layer, system/operation layer, data/information layer and behavior layer. Conventional methods mainly guarantee the security of IoT from the aspects of the system and data layers, and even the behavior layer, and propose corresponding protection strategies. However, the security risks from

the device/physical layer have been overlooked. A hostile hacker may also launch attacks from the underlying hardware of the devices. At this time, software security may be ineffective. To effectively mitigate these physical threats, a variety of protection mechanisms have been invented and employed, including secret key assignment and cryptographic technology [9–11]. However, these IoT devices usually have limited hardware resources and poor self-protection capabilities, whereas classical encryption solutions are often resource-/time-intensive [12–15], making them unavailable for IoT applications. In view of this, it is both necessary and urgent to develop lightweight solutions to ensure the security of IoT devices.

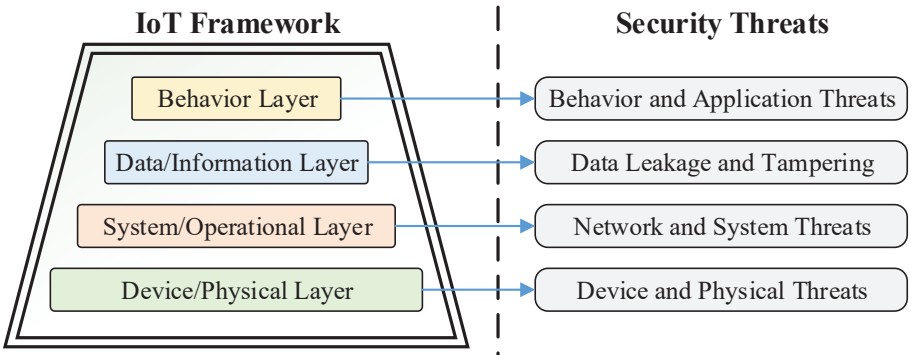

**Figure 1.** Security threats model of IoT framework.

A physically unclonable function (PUF) is a low-cost, hardware-based alternative solution to provide secure key generation and lightweight authentication for physical devices [14]. It exploits the intrinsic physical randomness of the chip manufacturing process to generate unclonable and unpredictable "hardware fingerprinting", or to generate a large amount of challenge–response pairs (CRPs) to facilitate device authentication and prevent various malicious attacks [16–18]. Compared to traditional encryption methods, the main advantages of PUFs are their uniqueness, unclonability and compatibility with IoT devices containing limited computing resources. As the PUF responses are extracted from the minor random variations of the internal chip, even if two PUFs have exactly the same structure, the response values are also different [15]. Moreover, due to the characteristics of being simple in structure, low in cost and fast in authentication, an individual PUF entity can be easily embedded into IoT devices as a hardware-based primitive for secure various applications, such as IP protection [3,19], authentication protocols [5,20], key sharing [6] and privacy-aware identification [21].

Since Pappu et al. [22] first presented the concept of PUF in 2002, many researchers have been committed to the study of PUFs and have presented a variety of different PUF architectures, including the static random access memory (SRAM) PUF [4,11], butterfly PUF [9,13], arbiter PUF [14], configurable ring oscillator (CRO) PUF [3,6], bistable-ring (BR) PUF [23], XOR-based reconfigurable BR (XRBR) PUF [15] and threshold implementation-based (TI) PUF [24]. Despite the fact that they all exhibit good performance with regard to their uniqueness and reliability, several limitations, such as their poor environmental stability, low applicability and complex operations, still more or less exist. In particular, the pico-PUF (PPUF) has proved to be a potential candidate for its simple structure and small environmental dependence [12]. Moreover, it can be applicable to all types of the field programmable gate array (FPGA) matrices [2,10], and requires a small time to produce output bits. However, as a weak PUF, the PPUF contains a limited CRP space (in many cases, only one), and the number of response bits is proportional to the PUF size [23,25], resulting in low hardware resource utilization. In fact, PUFs should have the highest possible bits per cost (BPC) ratio; that is, a PUF instance should produce a larger CRP space, whereas the overhead should be as low as possible. However, current studies on PUFs primarily focus on improving their reliability, and many PUFs (including PPUF) have been reported in the literature to undergo significant hardware overhead [15]. Therefore, they may be

difficult to effectively implement on FPGA-based IoT devices [14,17]. Recently, several improved methods have been proposed to enhance their uniqueness and reliability [12,13], or to expand their CRP space [16]. This, however, still consumes excessive hardware/time resources and may not be feasible in practice, especially for IoT devices with limited hardware resources. Our goal is to solve these problems in order to increase the BPC ratio without compromising the PUF performance.

In this article, we present a new ultra-lightweight and reconfigurable PUF structure, namely, reconfigurable PPUF (RPPUF), to solve the above problems. Our RPPUF is built on the PPUF proposal of Gu et al. [12]. By incorporating configurable logic structures, one RPPUF design can be configured into multiple different PPUF instances through the control input signals *K*, thereby scaling up the CRP space and increasing the efficiency of the hardware overhead. In particular, the main contributions of this article are as follows.

(1) A novel and ultra-lightweight RPPUF has been proposed;
(2) By embedding a configurable logic structure, the hardware resource utilization of PUF can be improved significantly. To the best of our knowledge, this is the first study to focus on the practical application of PUF architectures;
(3) A new performance measure, i.e., BPC, has been introduced to evaluate the hardware resource utilization of PUFs;
(4) We conduct the experimental RPPUF hardware implemented on FPGAs. The results show that, under the conditions of a similar PUF performance and same CRP space size, the resource overhead is dramatically reduced, making it more practical in IoT applications.

The remainder of this article is organized as follows. Section 2 elaborates on the related work of PUFs. Section 3 introduces the proposed RPPUF architecture. The experimental results and analysis are given in Section 4. Finally, a conclusion is drawn in Section 5.

## 2. Related Work

### 2.1. Traditional PUF Structures

Generally, the current mainstream silicon PUF schemes can be roughly classified into two categories, namely, memory-based PUFs and delay-based PUFs, depending on the different sources of uncontrollable manufacturing process variations [2]. Since PUFs are generated by using the inherent characteristics of hardware, a research trend is to explore the already deployed resources of the devices to generate unique fingerprints [3]. For example, the fingerprints can be extracted from the devices' memory. These memory-based PUFs usually utilize the stable states to which the memory units tend after being powered up as the entropy source to generate PUF responses [5,13]. For example, SRAM PUF [26], butterfly PUF [9] and flip-flop PUF [10] are typical memory-based PUFs. In 2007, Guajardo et al. [14] noticed that the SRAM cells startup with a random value of 0 or 1. The random values can be exploited to extract a unique digital signature. Since not all devices support SRAM, it may not be applicable in practice. To solve this problem, Kumar et al. [9] proposed a butterfly PUF, which uses a latch structure to replace the inverter on the cross-coupling circuit. However, the effectiveness needs to be further discussion and improved. Moreover, memory-based PUFs based on other memory technologies, such as DRAM and RRAM, have also been proposed [11,27]. However, like SRAM PUFs, the feasibility of DRAM PUFs in a given system also requires specific conditions to generate digital signatures; for example, every device needs to have DRAM, and an extra non-volatile memory (NVM) is required to store the secret keys generated by memory-based PUFs.

To overcome the above constraints, some researchers attempt to deploy new circuit modules in electronic devices and apply them as PUF units. This type of PUF typically utilizes the delay deviation between two paths in a circuit as the entropy source to extract fingerprints; they are named delay-based PUFs. For example, arbiter PUFs and RO PUFs are typical delay-based PUFs [14,15]. Arbiter PUFs generate a 1-bit PUF response by comparing the delay difference between parallel multiplexer chains [18]. The excitations *C* are the selection inputs of two MUX chains. Each excitation input bit $C_i$ determines

whether the step signal is transmitted in parallel or across. However, since the parallel two multiplexer chains in an arbiter PUF require symmetric routing, the performance is poor when implemented on FPGAs [2]. Another typical delay-based PUF is the RO PUF. This PUF employs the frequency difference among multiple ROs to generate random digital signatures. However, due to the impact of cascading, the RO PUFs have a limited number of CRPs and are not hardware-resource-efficient. In 2014, Gu et al. [12] presented a novel PUF architecture called PPUF. Though it is simple in structure, easy to implement and fast to respond, PPUF is a weak PUF with an even smaller number of CRPs, and still needs to be improved in terms ofits hardware resources.

### 2.2. Logic Reconfigurable PUF Structures

In order to provide a highly efficient utilization of hardware resources, some researchers have proposed the concept of logic reconfigurable PUF (LR PUF) architectures [15]. The configurable logic circuits or programmable delay lines (PDLs) are introduced into traditional arbiter and RO PUFs to produce a variety of the PUF variants. For example, Zhang et al. [6] presented a crossover RO PUF structure with higher flexibility. Multiple interstage crossing cells are inserted between every two adjacent stages of inverters of a classic *m*-stage RO PUF to logic configurability. The configurable information determines the propagation path of each step signal input. Liu et al. [15] proposed two reconfigurable PUF designs (XRRO PUF and XRBR PUF) based on the RO PUF and the BR PUF proposal of Chen et al. [23]. An XOR gate is a circuit that can be converted to a buffer or inverter by using one input as the control signal. Liu et al. utilize the XOR gates to replace the inverters in one signal RO to generate different RO instances. Therefore, the efficiencies of the PUFs increase. However, the PUF performance is not significantly improved and the hardware overhead still needs to be optimized.

Although LR PUFs can improve the PUF reliability and/or the number of response bits, they are rarely adopted in practice [28]. This is, in part, due to the fact that the reconfigurable logic may result in an increase in hardware resources overhead and operation complexity. In fact, in many IoT applications, the electronic devices, e.g., bluetooth headsets, electronic pens, smart bands and portable batteries, usually have very limited resources to implement generic lightweight hardware-based security primitives. Thus, these so-called low-cost LR PUF structures may not be appropriate for being deployed in these devices. Moreover, as delay-based strong PUFs, they have been demonstrated to be vulnerable to modeling attacks, though many solutions to alleviate this issue have been proposed in recent years [6,18,24,29]. At this point, weak PUFs are more advantageous because it is more costly for an adversary to model them [28,30]. However, the size of their CRP space is very limited and an extra NVM is also required to store the secret keys generated. This means that a new ultra-lightweight LR PUF architecture that fulfils the practical criteria of low-cost and more CRPs is both interesting and necessary. Therefore, in this article, we focus on the PPUF and how we can overcome these shortages by using reconfigurable logic. Furthermore, this is the novelty and motivation of our study.

## 3. The Reconfigurable Pico-PUF

This section depicts the overall architecture of the proposed RPPUF. Section 3.1 describes the overview of RPPUF. Section 3.2 introduces the operation process for bit generation. In Section 3.3, we present the characteristics of the proposed RPPUF.

### 3.1. General Description of RPPUF

The conventional PPUF circuit utilizes the delay deviations on two time delay paths to generate a random number [9,31]. Figure 2 shows the 1-bit circuit design cell of the PPUF architecture. Each delay path is implemented by a D-type flip-flop (FF) and a wire connecting FF and a NAND-based set-reset (SR) latch. The resulting state is intrinsically dependent on the fabrication process variations, corresponding to a stable *logic*-0 and *logic*-1 [2–4]. PPUF typically exploits this random variation as a source to extract unique

digital signatures. Such randomness, however, cannot guarantee the uniqueness and reliability of the responses simultaneously [16]. Although several methods have been proposed to enhance the performance of PPUF—for example, increasing the fan-out capability of the FF output to improve its reliability [13]—the problems of low hardware resource utilization and small CRP space have not been well resolved. The novel RPPUF presented in this article intends to create a strong and resource-efficient logic reconfigurable PUF scheme by assigning the control input signals *K[0: n-1]* to configurable logic circuits at every challenge.

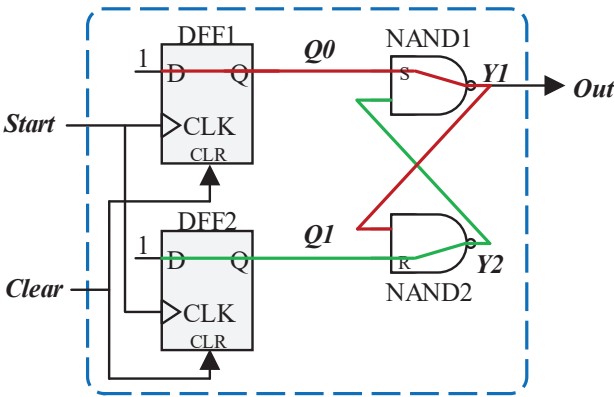

**Figure 2.** Circuit design of each pico-PUF architecture [12].

Figure 3 presents a schematic design of the proposed RPPUF circuit with two *configurable logic* structures. It is essentially a simple NAND-based SR latch with two FF structures and two *configurable logic* circuits connected before the SR latch. In the *configurable logic* circuits, a 2:1 multiplexer (MUX) unit is implemented at each stage to select one of the two inverters (INVs) as the delay element. Reconfiguration is reflected in the logic gates. A *configurable logic* with $n$ stages can be configured into $2^n$ different paths through the control input ports *K[0: n-1]*. Therefore, the proposed RPPUF can ideally produce a $2^{2n}$-number of possible different paths, i.e., exponential in $n$.

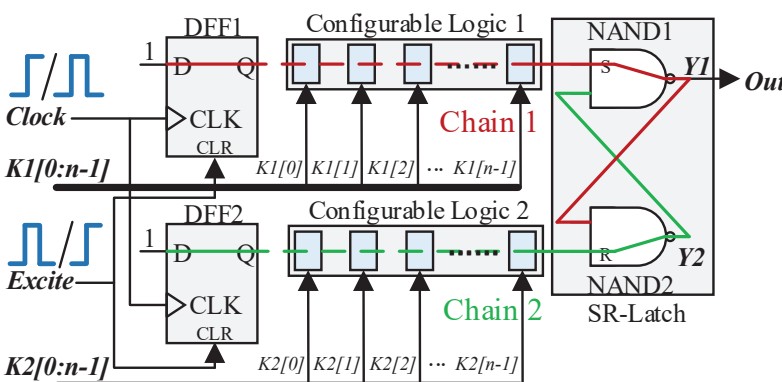

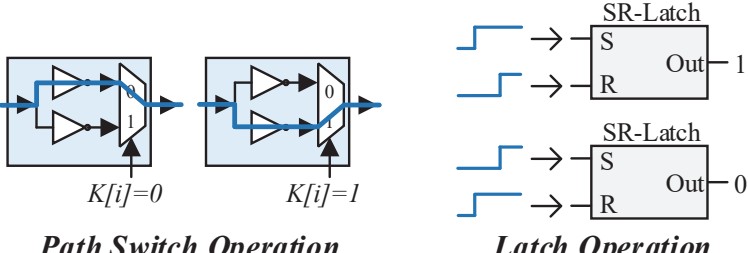

**Figure 3.** The proposed RPPUF design with configurable logic.

### 3.2. Operation Process for Bit Generation

To start the RPPUF operation, we first set the *excite* signal to *high* each time the challenges are valid to initialize the SR latch state; at this time, $Y1Y2 = 11$. Then, we reset *excite* to *low* and apply a *low-to-high* transition on the *clock*. The data $D$ in each FF circuit are transferred to the SR latch through the chain. Since the propagation delay between *Chain 1* (red line) and *Chain 2* (green line) may be slightly different due to the process variations in circuit parameters, the time for the data $D$ to reach the input ports $S$ and $R$ may also be different. This implies that the rising edge of data $D$ on one chain may come earlier than the other, which drives the SR latch to be biased to one of the two possible stable states, 0 or 1. In this case, the delay variation of the configurable logic circuits also participates in contention. The single response bit *out* generated through this simple comparison can be described by Equation (1). If $S$ is faster than $R$, $Y1Y2 = 01$, i.e., the response bit *out* samples 1; otherwise, $Y1Y2 = 10$, i.e., the *out* becomes 0.

$$Out = \begin{cases} 1 & (\text{i.e., } Y1Y2 = 01), \quad if \, \Delta S < \Delta R, \\ 0 & (\text{i.e., } Y1Y2 = 10), \quad otherwise. \end{cases} \tag{1}$$

Note that the number of stages in a configurable logic circuit can be either odd or even. For an RPPUF with $n$ stages *(n is even)*, the operation process is described as above. However, for $n$-numbered stages *(n is odd)*, the operation may be different. Since there are an *odd* number of INVs in the chain, the actual values at the input ports $S$ and $R$ of the SR latch are *1* when *excite* is set to *high*. For this reason, we adjust the operation sequence of the *clock* and *excite* signals; that is, whenever the challenge signals are valid, we first apply a *low-to-high-to-low* transition on *clock*, and then we set *excite*. In particular, the specific operation process is shown in Figure 4.

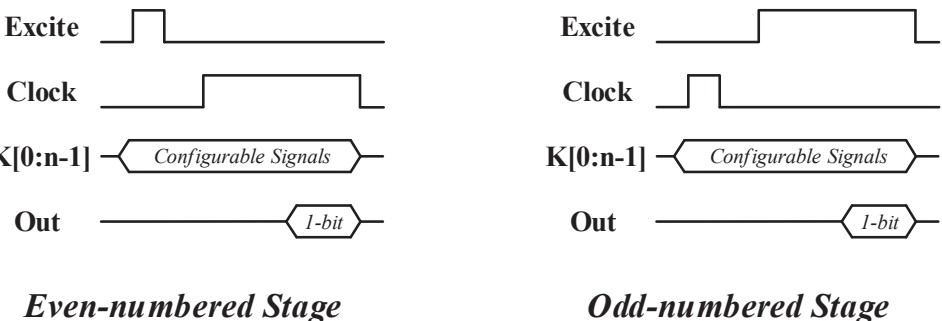

**Even-numbered Stage**        **Odd-numbered Stage**

**Figure 4.** The specific operation with different number of stages.

When implemented on FPGAs, only the two circuit elements (including logic and wires) that are compared directly require the identical structure and symmetrical layout manually, e.g., FF1-to-FF2 and NAND1-to-NAND2. Thus, the randomness in PUF responses can only be induced by the process variations [3,9,14]. This, to a certain extent, weakens the influences of the asymmetrical layout. Moreover, we also exploit the same configurable information values to the control inputs $K[0 : n-1]$ of the two configurable logic circuits to construct the identical paths each time the PUF responses are generated. As a result, the presented RPPUF scheme can, in fact, produce a $2^n$-bit output number.

### 3.3. Characteristics of RPPUFs

Several characteristics of the RPPUF designs are described as follows:

(1) It is resource-efficient. Despite the fact that the implementation complexities of the proposed RPPUF on FPGAs are increased slightly, it completely utilizes the configurability of logic structures to greatly reduce redundancy;

(2) It expands the CRP space. It expands the number of CRPs in a single PUF cell from 1 to $2^n$ by using the configurable information $K$. At this time, the format of CRPs has changed from *(C, R)* to *((C, K), R)*;

(3) The operation process for bit generation is simple and no error correction is required;

(4) It has a strong practicality. It can be effectively implemented and applied on FPGA-based IoT devices, with minimal hardware resources overhead and no complex circuit-level manipulation;

(5) However, as PPUF changes from weak to "strong", it is itself vulnerable to security threats, such as machine learning (ML)-based attacks accordingly;

(6) Furthermore, the requirement to balance the FPGA symmetrical routing and layout is more pertinent.

## 4. Experimental Evaluation

To demonstrate the proposed RPPUF design, we implement and evaluate it experimentally on FPGA microboards, which comprise a Xilinx Spartan-6 XC6SLX25 device (using 45 nm technology) to emulate the ASIC scenario. A 128-bit RPPUF design, in which each *configurable logic* contains seven stages, has been programmed into 10 identical Xilinx Spartan-6 LX25 FPGA microboards to generate a total of 10 individual implementations for testing. The PUF response bits are collected by an ATMEGA 2560 microboard, and are subsequently sent to the host computer through the USB-UART serial communication interface. In particular, Figure 5 presents the hardware experimental platform utilized to implement the proposed RPPUF design. It is specifically implemented as a hard macro, and the floor plan location is set by declaring location (LOC) constraints using Xilinx's unified constraints format (UCF) file [12].

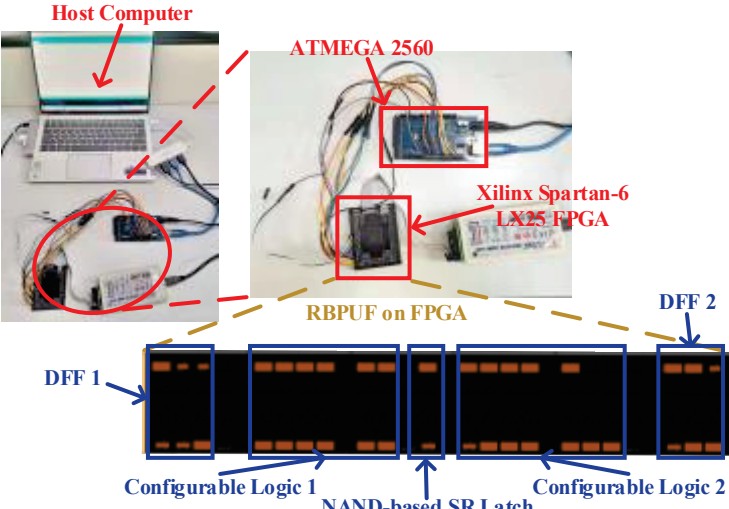

**Figure 5.** Hardware experimental platform and floorplaning of RPPUF implemented on Xilinx Spartan-6 FPGAs.

In this work, we utilize *five* important metrics to quantify the performance of the RPPUF circuit, i.e., effectiveness, uniqueness, reliability, uniformity and hardware overhead. In addition, we also implement the 128-bit PPUF [12] and reliability-improved butterfly PUF (4DBPUF) [13] designs on the same hardware development platform, and compare their performance to our proposed RPPUF design using these *five* important metrics to comprehensively evaluate and prove the their practicability.

### 4.1. Effectiveness

Effectiveness refers to the ability of the hardware inherent differences to affect the random response of PUFs [3]; *that is*, given the PUF entity $F$ and a challenge $C$, it can generate response output $R$; and when $C$ or the PUF location alters, $R$ will change accordingly. Since PPUF only contains one CRP, here, we change the configurable information $K$ of RPPUF. Figure 6 presents examples of the 128-bit response sequences generated by the RPPUF design under different excitation signals and PUF locations.

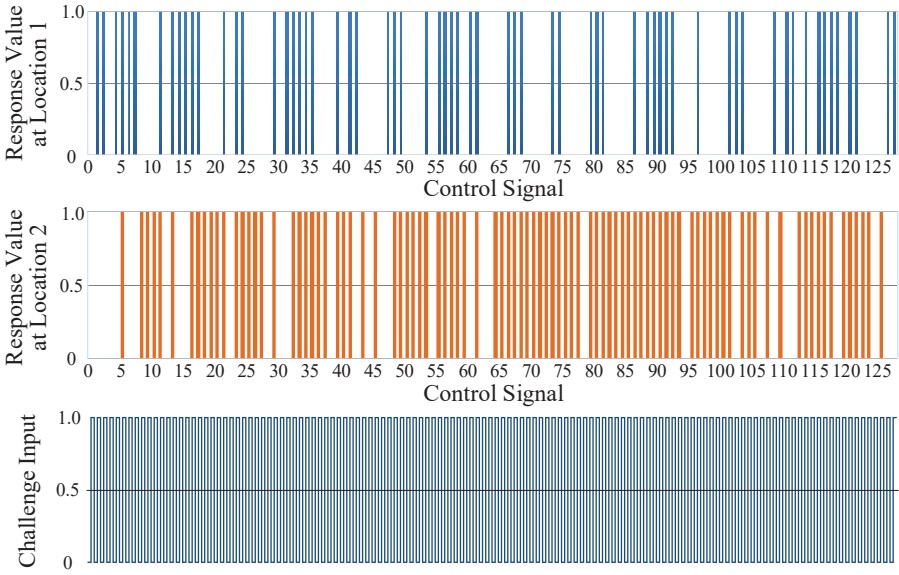

**Figure 6.** Samples of 128-bit digital signatures under different control signals and locations.

From Figure 6, we can conclude that, with the control signal *K* changes, *R* will change when *C* is applied, and the alternation of the PUF location in FPGAs will also result in a corresponding change in *R* under the same control signal sequences *K*, which proves the effectiveness of the RPPUF design.

*4.2. Uniqueness*

The uniqueness estimates the inter-chip variation by evaluating the PUF's ability to differentiate between *k* different devices [12]. This variation is quantified by the average inter-chip hamming distances (HDs) of the PUF responses over a group of devices [2,3]. For a pair of PUF entities, $F_i$ and $F_j$ ($i \neq j$), which both generate *n*-bit digital signatures, the uniqueness, representing the average inter-chip HD, can be defined as:

$$Uniqueness = \frac{1}{k(k-1)} \sum_{i=1}^{k-1} \sum_{j=i+1}^{k} \frac{HD(R_i, R_j)}{n} \times 100\% \quad (2)$$

where *k* represents the number of different devices; $R_i$ and $R_j$ are the *n*-bit responses extracted from two PUF designs, $F_i$ and $F_j$, respectively, under the same challenge *C*; and $HD(R_i, R_j)$ is the inter-chip HD between two different responses, which can be calculated as follows:

$$HD(R_i, R_j) = \sum_{l=1}^{n} R_i[l] \oplus R_j[l] \quad (3)$$

Figure 7 presents a histogram of the uniqueness distribution for the proposed RP-PUF design. Ideally, the average inter-chip HD of a PUF design that generates a 128-bit digital signature should converge to 64, which indicates better quality uniqueness. As can be seen from Figure 7, the average inter-chip HDs for PPUF, 4DBPUF and the proposed RPPUF design are 53.33, 53.51 and 58.62, respectively. The result of RPPUF has the smallest deviation from the ideal value of 64, with a difference of Δ = 5.38. Moreover, the quantitative comparison of uniqueness results between them is presented in Table 1, wherein the uniqueness of the proposed RPPUF circuit achieves an empirical mean of 45.80% on the Spartan-6 FPGAs, whereas the uniqueness value of 4DBPUF is approximated to that of PPUF, which is 41.67% and 41.81%, respectively. Compared to the studies on PPUF [12] and 4DBPUF [13], the RPPUF method increases the uniqueness by 4.13% and 3.99%, respectively, which is more concentrated on the expected ideal value, i.e.,

50%. This demonstrates that the proposed RPPUF circuit design has a better capability to differentiate between different devices.

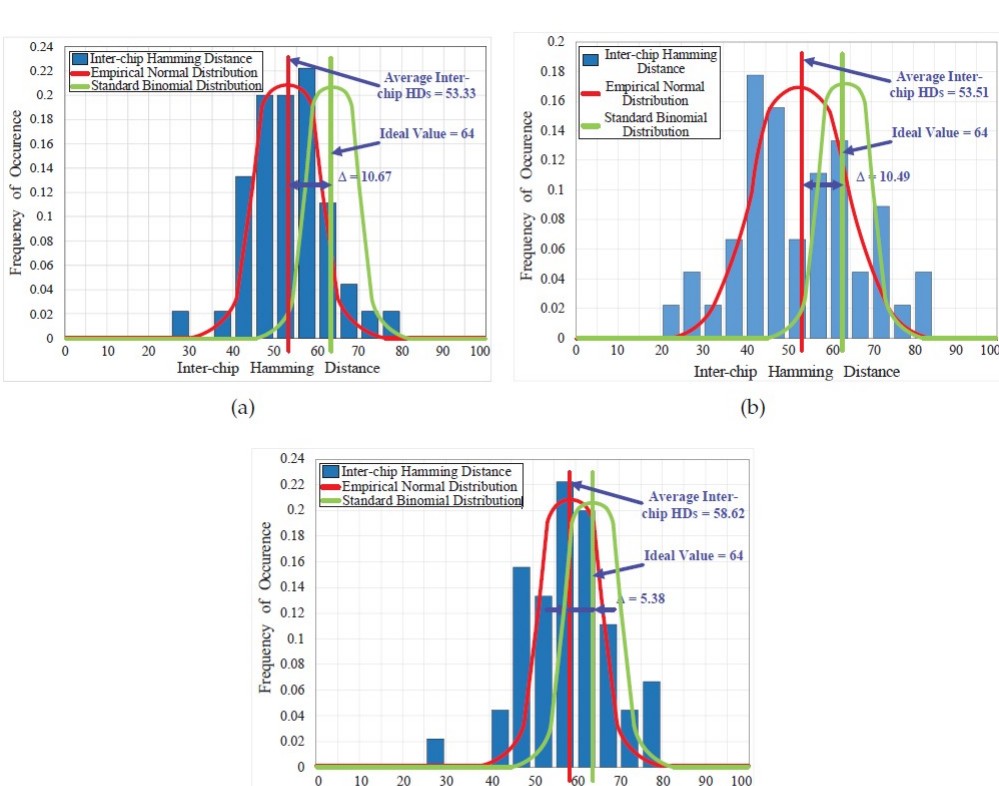

**Figure 7.** Distribution results of inter-chip HDs for (**a**) PPUF, (**b**) 4DBPUF and (**c**) RPPUF.

**Table 1.** Uniqueness comparison of RPPUF and other PPUFs.

|            | PPUF [12] | 4DBPUF [13] | RPPUF      | Ideal Value |
| ---------- | --------- | ----------- | ---------- | ----------- |
| Uniqueness | 41.67%    | 41.81%      | **45.80%** | 50%         |

*4.3. Reliability*

Reliability indicates that the PUF responses should remain consistent/stable at different environmental conditions [2]. Ideally, the responses reproduced by a given PUF circuit to the same challenge are not expected to contain any deviations. However, the environmental condition (such as the ambient temperature and power supply voltage) variations will induce noise in the PUF responses, which may result in one or more response bits to flip. Therefore, reliability is utilized to evaluate this difference in the PUF response. For a device *i*, the reliability can be quantified by using the average intra-chip HDs of the PUF response samples generated under different environmental conditions, which is defined as follows:

$$Intra\text{-}chip\ HD = \frac{1}{s}\sum_{t=1}^{s}\frac{HD(R_i, R_{i,t})}{n}\times 100\% \tag{4}$$

where *s* represents the number of response samples, $R_i$ is the baseline *n*-bit reference response extracted from a PUF design $F_i$ when the challenge *C* is applied under nominal operating conditions, $R_{i,t}$ is the *t*-th response sample reproduced under different operating conditions and $HD(R_i, R_{i,t})$ is the intra-chip HD between $R_i$ and $R_{i,t}$. In practice, the percentage figure of merit for reliability can be calculated as follows:

$$Reliability = 1 - Intra\text{-}chip\ HD \tag{5}$$

Here, we will measure the reliability of the proposed RPPUF circuit from two aspects, i.e., temperature and voltage. A 128-bit reference response is firstly generated from a device under normal operating conditions; *that is*, at a 20 °C room temperature and with a 1.2 V normal core supply voltage. Then, the operating temperature is varied from 20 °C to 80 °C (10 °C in each step) using a convection thermal incubator, while the core supply voltage is changed from 0.85 V to 1.35 V (0.5 V in each step) using a DC-regulated power supply. This roughly covers the permitted operating condition range of the FPGA. After that, the reference response is compared with the other response samples reproduced under various operating conditions to calculate the intra-chip HDs.

Figures 8 and 9 show the histograms of the distribution results of the temperature and core supply voltage intra-chip HDs for the proposed RPPUF design. In the ideal case, the expected value should be close to 0. From Figure 8, the average temperature intra-chip HD value of the proposed RPPUF design is slightly lower than that of 4DBPUF, but both are greater than PPUF. This implies that the 4DBPUF and the proposed RPPUF have a better temperature stability. Subsequently, analyzing Figure 9, the average core supply voltage intra-chip HD for the proposed RPPUF design has a minimum deviation from the ideal value of 0, with a difference of $\Delta = 4.44$. However, the results of PPUF and 4DBPUF are less preferable: the difference values of $\Delta$ are 16.58 and 28.36, respectively. This indicates that the proposed RPPUF circuit design has a high level of performance in terms of its core supply voltage stability.

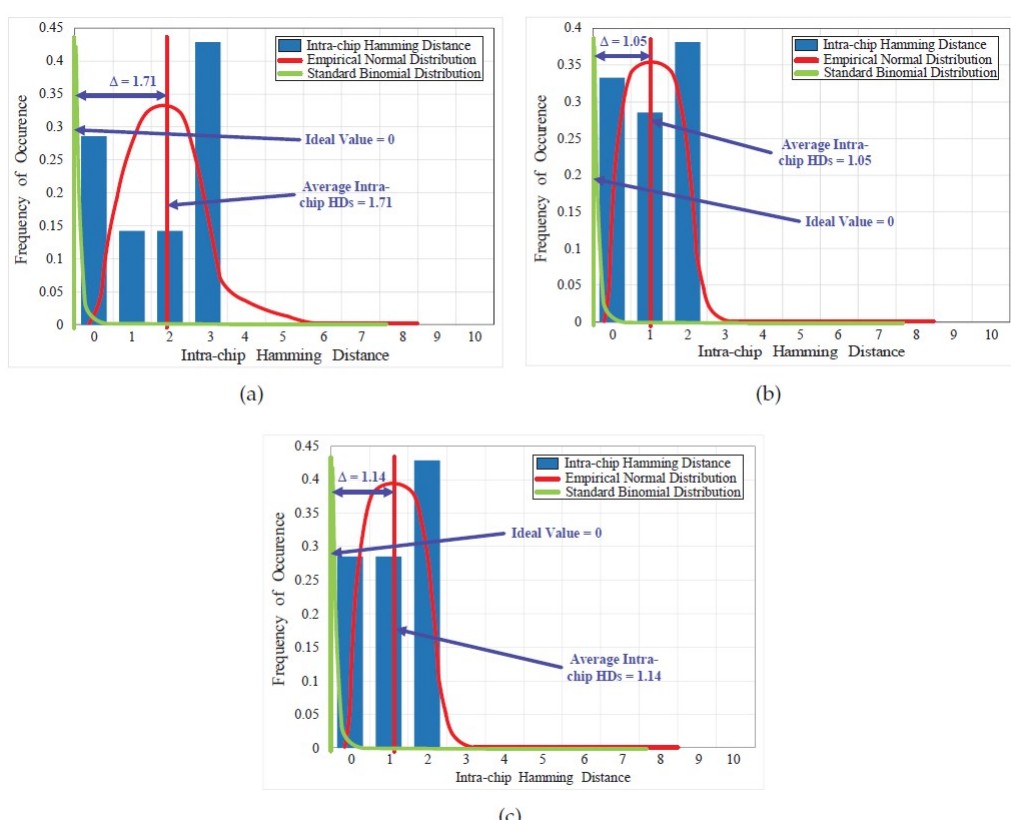

**Figure 8.** Distribution results of temperature intra-chip HDs for (**a**) PPUF, (**b**) 4DBPUF and (**c**) RPPUF.

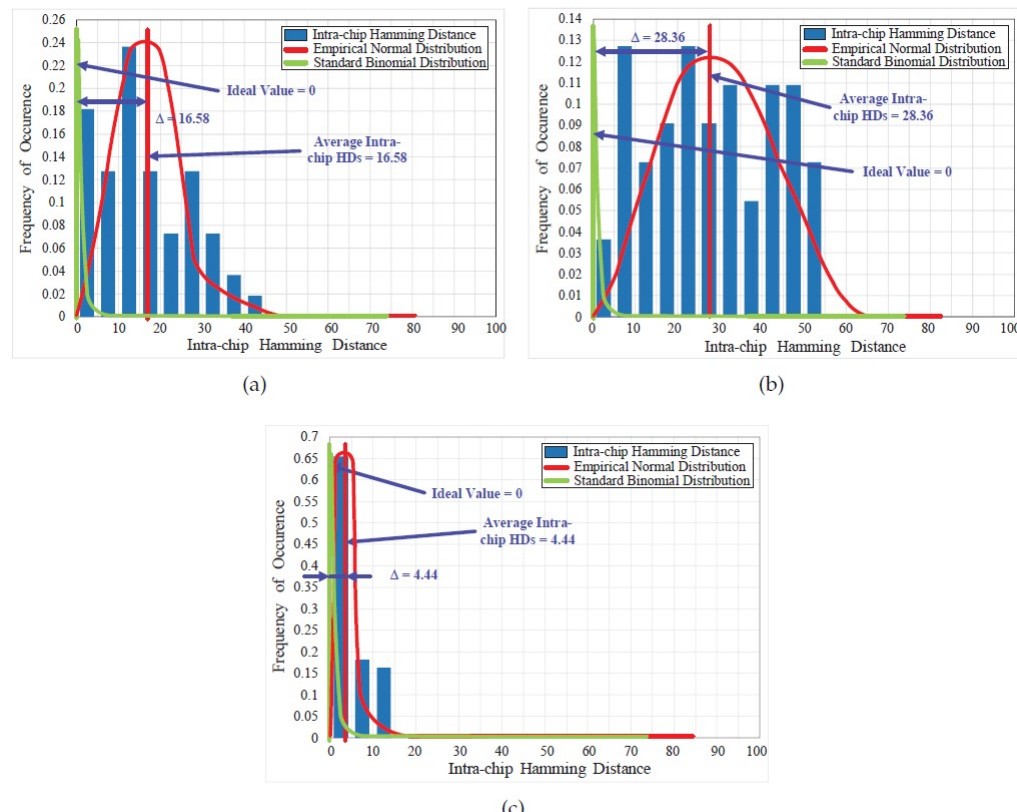

**Figure 9.** Distribution results of core supply voltage intra-chip HDs for (**a**) PPUF, (**b**) 4DBPUF and (**c**) RPPUF.

Moreover, Figure 10 describes the reliability results of PPUF, 4DBPUF and the proposed RPPUF circuit for the operating temperature and core supply voltage, and the quantitative comparison of reliability results between them is also presented in Table 2, wherein the reliability results of the proposed RPPUF circuit for temperature and voltage separately achieve the empirical means of 99.23% and 96.85% on the Spartan-6 FPGAs, whereas the values of the 4DBPUF and PPUF are 99.30% and 98.85%, and 79.87% and 88.22%, respectively. Compared to the studies on PPUF [12] and 4DBPUF [13], our RPPUF method may not be very obvious in increasing the temperature reliability. However, it significantly improves the voltage reliability by 8.63% and 16.98%, respectively, which is more focused on the desired ideal value, i.e., 100%. This demonstrates that the proposed RPPUF circuit design exhibits a high level of stability to resist environmental condition variations.

**Table 2.** Reliability comparison of RPPUF and other PPUFs.

|  | **PPUF [12]** | **4DBPUF [13]** | **RPPUF** | **Ideal Value** |
|---|---|---|---|---|
| **Temperature Reliability** | 98.85% | 99.30% | **99.23%** | 100% |
| **Voltage Reliability** | 88.22% | 79.87% | **96.85%** | 100% |

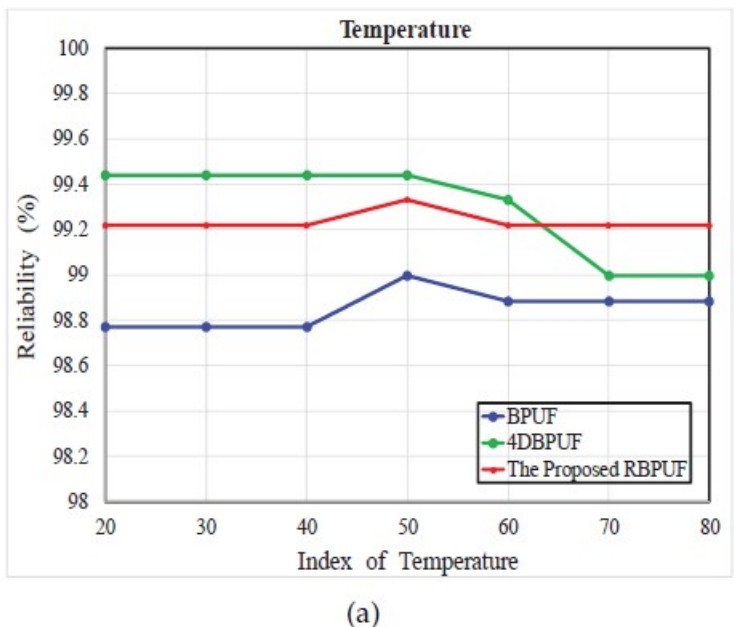

(a)

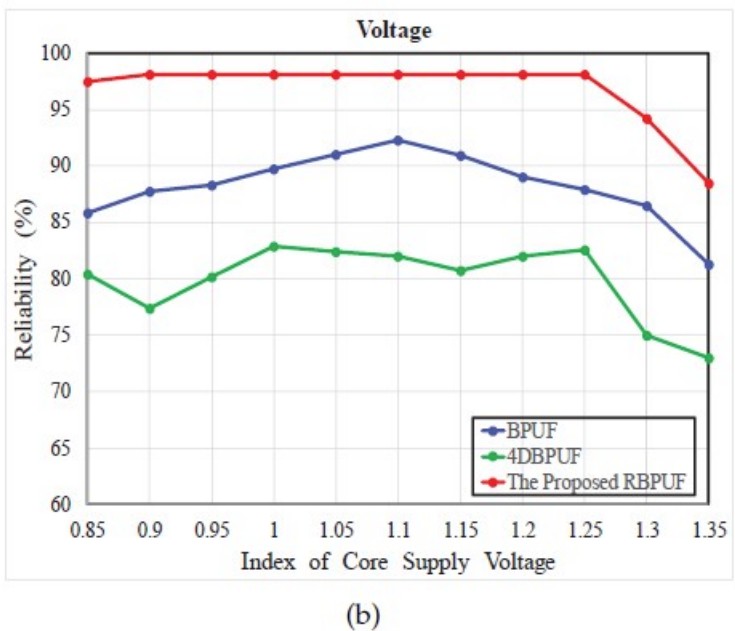

(b)

**Figure 10.** Reliability results over different environmental variations for (**a**) operating temperature and (**b**) core supply voltage.

### 4.4. Uniformity

Uniformity is utilized to measure the distribution proportion of "0" and "1" in a specified PUF response sequence [2,31]. This metric is related to the security of the PUF design [12]. For a truly random PUF signature, the probability for each bit value must be equal. When the response of a specific device is biased towards a particular value, the security will be affected to some extent. In particular, the hamming weight (HW) can be applied to estimate the uniformity of a PUF response, which is given as follows.

$$Uniformity = \frac{1}{n} \sum_{l=1}^{n} R_i[l] \times 100\% \tag{6}$$

where $R_i[l]$ is the $l$-th bit of an $n$-bit response extracted from a specific PUF design $i$.

In this work, we measure the uniformity of PPUF, 4DBPUF and the proposed RPPUF, and the proportional results of "0s" and "1s" in their response sequences are presented in Figure 11. It can be seen from Figure 11 that the average HW values observed for them are 0.546, 0.375 and 0.48, respectively. Therein, the results of PPUF and RPPUF are much closer to the desired value of 0.5 (the deviations $\Delta = 0.046$ and 0.02). However, the deviation for that of 4DBPUF circuit is larger. From further analysis of Figure 11, we conclude that the uniform distribution of PPUF is relatively more concentrated. This implies that the fluctuations in randomness of PPUF responses is the smallest, followed by RPPUF, and the worst is 4DBPUF. Moreover, the quantitative comparison of uniformity is given in Table 3. It should be 50% for uniformity. In particular, the uniformity of the proposed RPPUF circuit achieves an empirical mean of 48.0% on the Spartan-6 FPGAs. Compared to the studies on PPUF [12] and 4DBPUF [13], the RPPUF method increases the uniformity by 2.6% and 10.5%, respectively. This demonstrates that the signatures extracted from the proposed RPPUF circuit have a better quality of randomness.

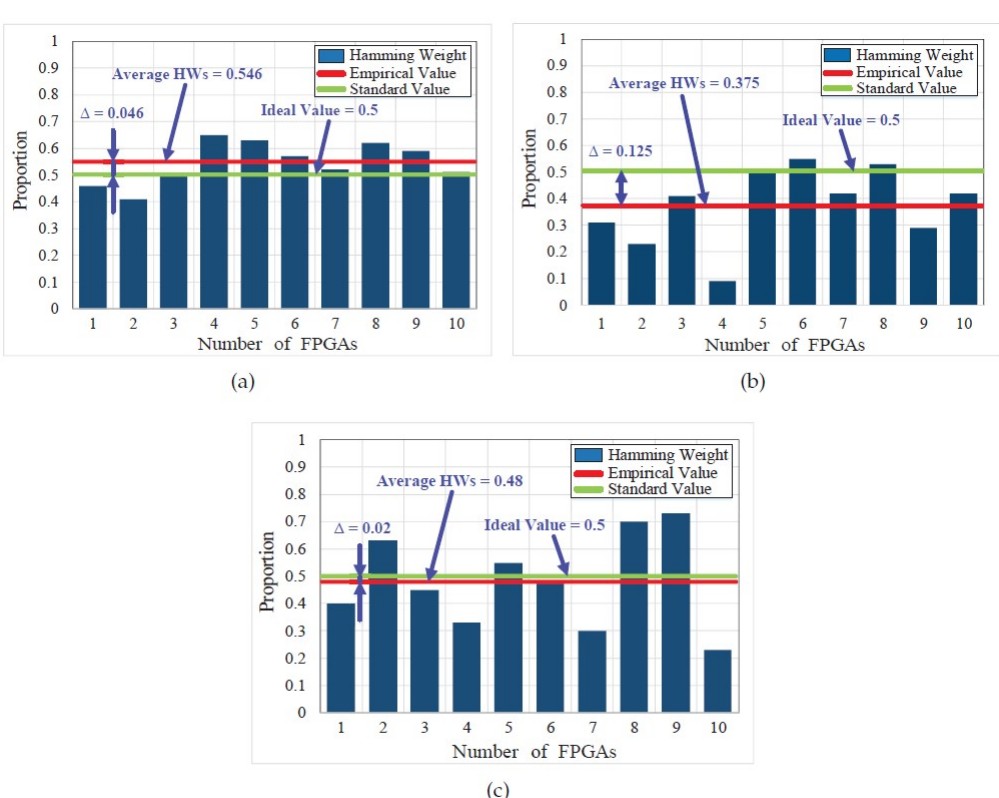

**Figure 11.** Proportional results of "0 s" and "1 s" in the responses for (**a**) PPUF, (**b**) 4DBPUF and (**c**) RPPUF.

**Table 3.** Uniformity comparison of RPPUF and other PPUFs.

|  | **PPUF [12]** | **4DBPUF [13]** | **RPPUF** | **Ideal Value** |
|---|---|---|---|---|
| **Uniformity** | 54.6% | 37.5% | **48.0%** | 50% |

### 4.5. Hardware Overhead

As a promising hardware-based security primitive, a low cost is a required property of PUF. In this work, we utilize a further metric, *bits per cost* (BPC), to evaluate the cost efficiency of a PUF implemented on FPGA [15]. The number of response bits generated in the unit hardware resource area is defined as BPC. The BPC can be utilized as a metric to compare the cost efficiency of different PUF circuits implemented on FPGAs. If a PUF design can use the same hardware resource to generate a longer length of response, it is considered to be a more cost-efficient design. Therefore, the higher the BPC, the higher the cost efficiency of the PUF circuit. Based on this description, BPC can be quantified as follows.

$$BPC = \frac{n}{A} \tag{7}$$

where $n$ is the length of a PUF response $R_i$, and $A$ is the total hardware resources overhead. For a PUF design implemented on FPGAs, a higher BPC value means a higher cost efficiency. The hardware overhead occupied by PPUF, 4DBPUF and the proposed RPPUF circuit on Xilinx Spartan-6 LX25 FPGAs in generating a 128-bit signature is shown in Table 4.

**Table 4.** Hardware overhead.

|  | PPUF [12] | 4DBPUF [13] | RPPUF |
| --- | --- | --- | --- |
| # of LUTs | 1835 | 3371 | 62 |
| # of Registers | 0 | 0 | 0 |

According to Table 4, 4DBPUF consumes the largest number of LUTs, which is 3371. This is primarily due to the fact that each 4DBPUF cell contains *four* FF structures, *two* NAND gates and *two* buffer units. Therefore, 4DBPUF has a higher hardware resource consumption than that of PPUF, whose cell only contains *two* FF structures. However, the number of LUTs that PPUF occupies is still large, at 1835. As expected, our proposed RPPUF circuit design only occupies 62 numbers of LUTs, accounting for 3.38% and 1.84% when compared to PPUF [12] and 4DBPUF [13]. This indicates that the RPPUF can dramatically reduce the hardware overhead by the percentages of 96.62% and 98.16%, respectively.

Next, we utilize the BPC to estimate the cost efficiency of PPUF, 4DBPUF and the proposed RPPUF. Since the signature generated by them are all 128 bits, $n$ is a constant value of 128. In particular, Figure 12 presents the BPC results of the proposed RPPUF and compares them to other PPUFs. The BPC result of the proposed RPPUF is significantly higher than that of PPUF and 4DBPUF under limited resources, which is 2.065. This implies that the hardware cost efficiency for the RPPUF implemented on Xilinx Spartan-6 LX25 FPGA fabrics in generating a 128-bit signature is higher. Therefore, it is more suitable as a low-overhead and hardware-based primitive for IoT devices. Compared to PPUF [12] and 4DBPUF [13], the hardware cost efficiency of the proposed RPPUF increased by 28.5 and 53.34 times, respectively.

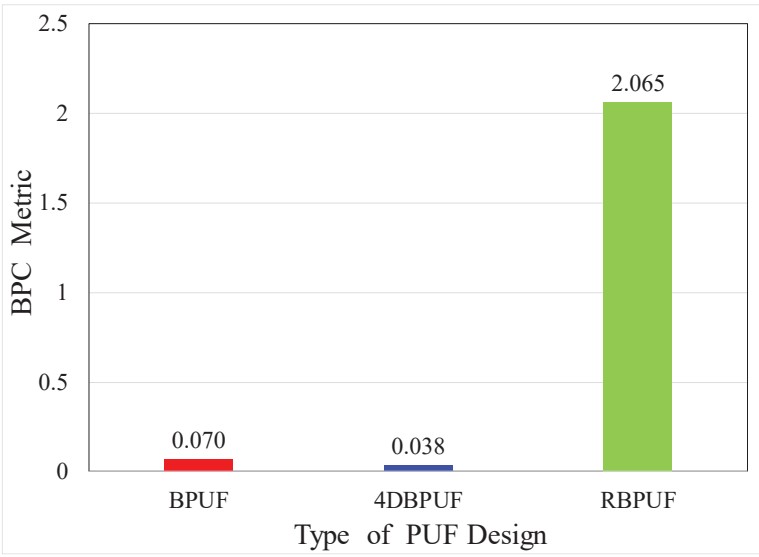

**Figure 12.** BPC results of PPUF, 4DBPUF, and RPPUF.

### 4.6. ML Attack Results

Ideally, PUFs are unpredictable, unclonable and tamper-evident [2]. However, several attack techniques on PUF properties have been reported recently that can successfully construct the software models for PUFs. Thereinto, machine learning (ML)-based attacks are the most efficient attack technique against strong PUFs [6]. Since the strong PUFs have a publicly accessible CRP interface, an adversary can collect a certain amount of CRPs from previous authentication sessions and train them using MLs to obtain an expected PUF model. As a result, it will accurately predict the future responses to any input challenges. For example, Zhang et al. has proven that MLs can successfully model the arbiter PUFs (APUFs) with prediction rates of up to 99% [2,3,6]. In this section, we will evaluate the capability of the proposed RPPUF to resist ML attacks.

To evaluate the ML resistance of the proposed RPPUF architecture, we utilize some widely used ML algorithms, e.g., logistic regression (LR), artificial neural network (ANN), support vector machine (SVM), decision tree (DT) and random forest (RF), in order to complete ML-based modeling attacks. These five ML models are implemented using open source Python packages *scikit-learn* and are learned with training CRP sample sets varying in size from 20,000 to 100,000. Table 5 presents the corresponding parameters of each ML-based attack. In particular, an ANN attack is performed using two hidden layers with 100 iterations at a learning rate of 0.001. Each hidden layer consists of 40 neurons. The specific parameter settings of other ML models are listed in Table 5. We predict the RPPUF response in 128-bit lengths using these ML attack models, and the prediction rates are shown in Figure 13.

From Figure 13, it is obvious that RPPUF exhibits a slightly better resistance to the LR attack when compared to other ML algorithms, and the prediction rate is between 48.75% and 67.77%. Similarly, the prediction rate of the SVM attack for the RPPUF structure is close to LR, and the maximum value for them is not yet beyond 68%. Since the ideal prediction rate of the ML attack-resistant strong PUF is 50%, our RPPUF is more resistant to LR and SVM attacks compared to the other three ML models. In contrast, APUF is almost powerless against SVM and LR attacks. This result illustrates that APUF and RPPUF have different resistances to LR and SVM attacks. Moreover, the prediction rates of DT and RF for RPPUF are very high, approaching or even reaching 100% in most cases. This means that, even though the size of CRP sample sets is less than 20,000, the proposed RPPUF design can be successfully predicted by DT and RF. Therefore, it is difficult for RPPUF to resist DT or RF attacks. This may be due to the better learning capabilities of DT and RF to learn the underlying law about the data with fewer training samples, thereby

enabling them to achieve a high accuracy. However, both APUF and RPPUF fail to resist the ANN attack.

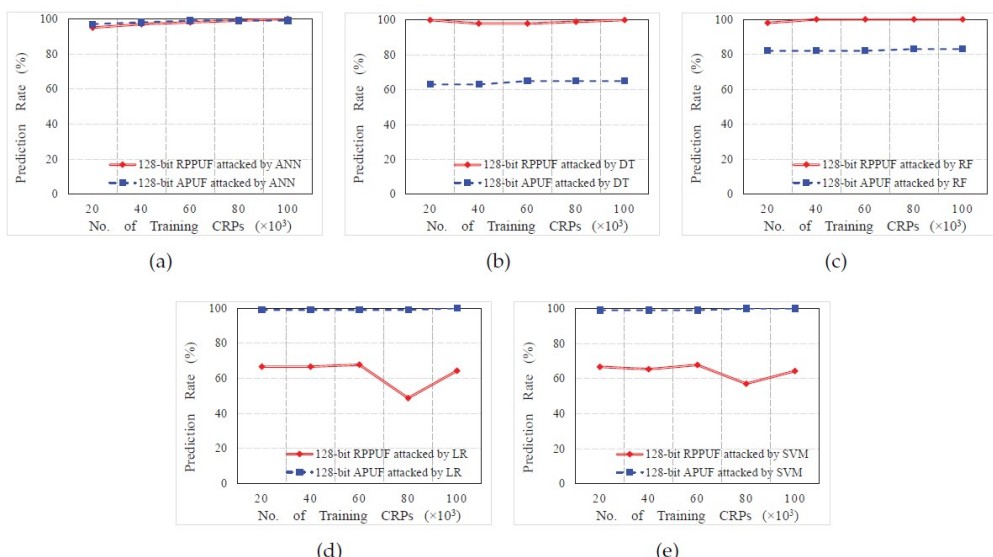

**Figure 13.** The prediction rates of various ML attacks on both RPPUF and APUF with 128-bit responses. (**a**) ANN, (**b**) DT, (**c**) RF, (**d**) LR and (**e**) SVM.

**Table 5.** Parameter setting for each ML attack models.

| Attack Model | Main Structure and Parameter |
| --- | --- |
| ANN | MLP, hidden_layer_sizes(40, 40), solver = 'adam', alpha = $10^{-4}$, activation = "relu", learning_rate_int = 0.001. |
| SVM | C_SVC, C=1.0, kernel = rbf, degree = 3, gamma = 'scale', coef0 = 0.0. |
| LR | penalty = 'l2', C=1.0, solver = 'liblinear', multi_class = 'ovr'. |
| DT | criterion = "gini", splitter = "best", min_samples_leaf = 1. |
| RF | n_estimators = 10, max_depth = 2, random_state = 0, min_samples_leaf = 1. |

The final prediction rates, *including* the average prediction rate (APR), maximum prediction rate (MaxPR) and minimum prediction rate (MinPR), and the training time required for the different ML attacks, are illustrated in Tables 6 and 7. It can be seen from Table 6 that the MinPR values for each size of training CRP sample sets are, relatively, not bad, basically remaining below 68%. Therefore, it is clear that the proposed RPPUF is resistant to one or some ML attacks under certain conditions. However, the APR value for each size of training CRP sample sets varies from 80.87% to 86.41%, whereas the MaxPR value of all training sample sets is 100%. From Table 7, it is evident that the ANN attack consumes the longest training time, although it exhibits good prediction results. However, the LR attack requires the least time for feature training, followed by DT. This means that the LR attack has the fastest execution efficiency. Since the DT attack contains a high predictive rate and a relatively lower training time for RPPUF, the attack success efficiency for DT is the highest.

In general, the ML resistance of the proposed RPPUF is relatively weak, especially for the ANN, DT and RF attacks. Thus, it needs to be further strengthened.

**Table 6.** Final prediction rates for different ML attacks.

| PUF Design | MinPR | MaxPR | APR | No. CRPs ($\times 10^3$) |
|---|---|---|---|---|
| | 66.66% | 100% | 85.66% | 20 |
| | 65.38% | 100% | 85.41% | 40 |
| RPPUF (128-bit) | 67.77% | 100% | 86.41% | 60 |
| | 48.75% | 100% | 80.87% | 80 |
| | 64.33% | 100% | 85.73% | 100 |

**Table 7.** Training time for different ML attacks.

| PUF Design | No. CRPs ($\times 10^3$) | Training Time | | | | |
|---|---|---|---|---|---|---|
| | | ANN | SVM | LR | DT | RF |
| | 20 | 80.57 s | 0.20 s | 37 ms | 88 ms | 0.80 s |
| | 40 | 155.04 s | 0.38 s | 62 ms | 122 ms | 0.82 s |
| RPPUF (128-bit) | 60 | 236.55 s | 1.18 s | 84 ms | 129 ms | 1.54 s |
| | 80 | 319.64 s | 1.31 s | 114 ms | 148 ms | 2.09 s |
| | 100 | 394.93 s | 1.85 s | 147 ms | 250 ms | 2.51 s |

*4.7. Comparison with State of the Art*

In this section, a performance comparison of the proposed RPPUF structure against the state-of-the-arts has been performed. Table 8 presents the summary of this comparison, which reports the overview of the expected uniqueness, reliability and uniformity results between the PUF architectures, including the proposed RPPUF, PPUF and improved butterfly PUF, as well as the previously proposed PUF designs. It should be noted that part of the data in Table 8 are from the original literature.

**Table 8.** A comparison of performance measures of different PUFs.

| PUF Design | Year | Type | Uniqueness | Reliability | | Uniformity |
|---|---|---|---|---|---|---|
| **Our RPPUF** [s] | 2021 | Strong | 45.80% | 99.23% [t] | 96.85% [v] | 48.0% |
| PPUF [12] | 2017 | Weak | 49.90% | 94.53% | | - |
| PPUF [s] [32] | 2014 | Weak | 41.67% | 98.85% [t] | 88.22% [v] | 54.6% |
| 4DBPUF [s] [13] | 2017 | Weak | 41.81% | 99.30% [t] | 79.87% [v] | 37.5% |
| DD-PUF [33] | 2021 | Weak | 49.28% | 98.37% | | 48.47% |
| DD-PUF [33] | 2021 | Weak | 49.48% | 98.33% | | 50.59% |
| Lattice PUF [34] | 2020 | Strong | 50.0% | 98.74% | | 49.98% |
| FF-APUF [35] | 2019 | Strong | 41.53% | 97.10% [t] | 93.90% [v] | - |
| I-APUF [35] | 2019 | Strong | 19.46% | - | 97.03% [v] | - |
| FD-APUF [36] | 2005 | Strong | 38.0% | 90.2% | | - |

[s]—data extracted by ourself. [v]—under supply voltage variation. [t]—under temperature variation.

From Table 8, it can be seen that these PUF architectures, except lattice PUF, have a delay-based design. The performance results for all PUFs are good, but several index values for 4DBPUF [13], improved APUF (I-APUF) [35] and feed-forward APUF (FD-APUF) [36] are relatively low. In fact, delay difference PUF (DD-PUF) [35], FF-based APUF (FF-APUF) [33] and our RPPUF are the results of different improvements to PPUF [12], which

is intended to increase the hardware resource efficiency without sacrificing performance; inevitably, they have similar structures. To the best of our knowledge, the proposed RPPUF improves the structure of PPUF by using *configurable logic* circuits, and the performance measures are very close to ideal values. This illustrates the performance effectiveness of our approach. Furthermore, the fuzzy extraction and post-processing strategies can be used to further optimize their performance [37,38].

Moreover, a comparison of the resource efficiency for different PUFs is shown in Table 9. It is evident from Table 9 that most PUF methods, with the exception of 4DBPUF [13], have a relatively close hardware resource efficiency. In particular, our proposed RPPUF design can generate a 128-bit response sequence with only eight configurable logic blocks (CLBs); hence, 16 Slices. Compared to other previously published methods, our method can achieve a remarkable improvement in resource utilization while maintaining the PUF performance. Therefore, this demonstrates the resource efficiency of our RPPUF design. In addition, we also summarize the number of input challenge bits required to produce 1 bit of the PUF response, which is abbreviated as NCBPR. Since PPUF, 4DBPUF and DD-PUF are weak PUFs (see Table 8), their NCBPR value is 1. As can be seen from Table 9, among strong PUFs, RPPUF has the smallest NCBPR value, which is 7. This indicates that, although the CRP space of RPPUF is expanded when compared to that of PPUF, it is still limited. Therefore, RPPUF has a high probability of being successfully predicted.

Furthermore, the comparison of security analyses for different PUF designs is provided in Table 10. Since ML attacks are ineffective against weak PUFs, Table 10 only provides the results of strong PUFs. As can be seen from Table 10, apart from traditional ML algorithms, several advanced ML techniques, *such as* deep neural networks (DNN), evolution strategies (ES) and covariance matrix adaptation ES (CMA-ES), can also be utilized to launch a modeling attack on PUFs and exhibit a better performance. For lattice PUF [34], the attack model is DNN, whereas for FF-APUF [35], the attack model is CMA-ES. At this time, the APR values for them are around 50.17% and 49.56%, respectively. This illustrates that the lattice PUF and FF-APUF are more resistant to these advanced ML attacks. However, the FD-APUF can achieve APR values of more than 86.8% by using ANN and ES. Therefore, it can easily predict the response sequences. To compare with the previously proposed approaches, we present the results of the proposed RPPUF, which is attacked by SVM, ANN, LR, DT and RF. It is evident that the APR of the RPPUF can achieve prediction rate values within 62.5–64.5% by applying SVM and LR. However, the attack results for ANN, DT and RF are over 97.8%, 99% and 99.6%, respectively. All of these results indicate that our RPPUF demonstrates a similar ML resistance to the APUF, I-APUF and FD-APUF, but is worse than the FF-APUF and lattice PUF design.

**Table 9.** A comparison of the resource efficiency of different PUFs.

| PUF Design | NCBPR | Response | Device | Resource | CLB/Bit | Slice/Bit |
|---|---|---|---|---|---|---|
| **Our RPPUF** [s] | 7 | 128 | Spartan-6 | 16 Slices | 1/16 | 1/8 |
| PPUF [12] | 1 | 128 | Artix-7 | 128 Slices | 1/1 | 1/1 |
| PPUF [s] [32] | 1 | 128 | Spartan-6 | 256 Slices | 1/1 | 2/1 |
| 4DBPUF [s] [13] | 1 | 128 | Spartan-6 | 640 Slices | 5/4 | 5/2 |
| DD-PUF [33] | 1 | 128 | Spartan-6 | 64 Slices | 1/4 | 1/2 |
| DD-PUF [33] | 1 | 128 | Artix-7 | 64 Slices | 1/4 | 1/2 |
| Lattice PUF [34] | 264 | 100 | Spartan-6 | 45 Slices | 1/4 | 1/2 |
| FF-APUF [35] | 192 | 64 | Artix-7 | 128 Slices | 1/1 | 2/1 |
| I-APUF [35] | 192 | 64 | Artix-7 | 128 Slices | 1/1 | 2/1 |
| FD-APUF [36] | 64 | - | TSMC 180nm | - | - | - |

[s]—data extracted by ourself.

**Table 10.** A comparison of machine learning attacks of different PUFs.

| PUF Design | Response | No. CRPs ($\times 10^3$) | ARP | Attack Model |
|---|---|---|---|---|
| **Our RPPUF** [s] | 128-bit | 20–100 | 97.80% | ANN |
| | | | 64.24% | SVM |
| | | | 62.83% | LR |
| | | | 99% | DT |
| | | | 99.6% | RF |
| Lattice PUF [34] | 100-bit | 1–1000 | 50.17% | DNN |
| FF-APUF [35] | 32-bit | 1–10 | 49.56% | CMA-ES |
| I-APUF [35] | 32-bit | 1–10 | 74.43% | CMA-ES |
| FD-APUF [36] | 64-bit | 50 | 98.66% | ES [39] |
| | 64-bit | 10–200 | 86.85% | ANN [40] |
| | 128-bit | 50 | 98.32% | ES [39] |
| | 128-bit | 20–200 | 87.83% | ANN [40] |

[s]—data extracted by ourself.

## 5. Conclusions

In this article, we present a novel RPPUF circuit in order to obtain strong and reconfigurable PUF characteristics. The reconfigurability is reflected in the logic gates. By adding *configurable logic* structures, we modify the conventional PPUF from being weak to being a "strong" PUF. Therefore, more CRPs will be generated in one single RPPUF design due to the *configurable logic* circuits with different control input signals. This, at the same time, increases the efficiency of the hardware cost, making it more practical in IoT applications. The proposed RPPUF has been implemented on Xilinx Spartan-6 FPGAs. Experimental results demonstrate that the proposed RPPUF is slightly better than PPUF and 4DBPUF in terms of its uniqueness, temperature reliability and uniformity, with values of 45.80%, 99.23% and 48.0%. Moreover, it achieves a much better performance in terms of its voltage reliability and BPC metrics, with values of 96.85% and 2.065. This indicates that the proposed RPPUF circuit is very efficient, secure and practical, with a high resource efficiency and performance; thus, it is more suitable for effective application in resource-constrained IoT devices.

Furthermore, the digital signatures generated by PUF architectures should be data-independent, making them unpredictable. However, the response strings of the proposed RPPUFs have a certain data dependency, i.e., some bits of a PUF response always have fixed values. Furthermore, the RPPUF may be not resistant to ML-based attacks, since they can be formulated as a linear additive delay model. Therefore, future work will include decreasing the data dependency of PUF responses and enhancing the security against ML-based modeling attacks.

**Author Contributions:** Conceptualization, Q.W. and Z.H.; methodology, Z.H. and Z.L.; software, Z.L. and L.L.; validation, Z.H. and Z.L.; writing—original draft preparation, L.L.; writing—review and editing, Z.H. and Y.C.; supervision, Q.W. and X.J.; project administration, Q.W.; funding acquisition, Q.W. All authors have read and agreed to the published version of the manuscript.

**Funding:** This work is supported by the National Natural Science Foundation of China (Nos. 61972302 and 61962019) and the Group Intelligence Behavior Analysis-based Cultural Material Identification and Digital Product Development & Reuse (2019YFB1406402), and is partly supported by the Shannxi Key Technology R&D Program (Nos. 2021ZDLGY07-01 and 2021ZDLGY07-04), and is partly supported by the Key Laboratory of Smart Human–Computer Interaction and Wearable Technology of Shaanxi Province.

**Acknowledgments:** The authors would like to thank the Assistant Chongyan Gu (Queen's University Belfast, QUB), Jiliang Zhang (Hunan University), and Lecturer Gang Li (Wenzhou University) for their suggestion. Also, thanks for the help of Postgraduate Junjie Wang (Xidian University) in the experiments.

**Conflicts of Interest:** The authors declare no conflict of interest.

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
