# Peer review of "RPPUF: An Ultra-Lightweight Reconfigurable Pico-Physically Unclonable Function for Resource-Constrained IoT Devices"

_electronics, doi:10.3390/electronics10233039_

Round 1
Reviewer 1 Report
The authors presented a design of reconfigurable Pico-PUF for resource-constrained IoT devices such as FPGAs. The authors claim to increase the CRP space from 1 (Pico-PUF) to 2^n. However, we found that the proposed design is not a novel one and does not show greatness than arbiter delay PUF. We also do not see any evidence or test result to show the machine learning resistance of the proposed PUF. Compared to existing strong PUF designs, the proposed PUF does not show advantages apart from reliability. In conclusion, I suggest rejecting this paper.
- The authors provide a detailed analysis of the reliability and uniformity, which looks reasonable. However, we are more interested in how to improve the reliability of the PUF with either hardware or software solution. And we also want to see analysis of the unpredictability to prove or explain why the proposed PUF has better resistance to machine learning attacks. It does not make sence to compare the machine learning resistance with 1-CRP Pico-PUF. There has already been works to show provable resistance to machine learning attacks [1][2]. So the authors need to either prove their high resistance or explain why the proposed PUF has advantage over these two PUFs.
- From the description of the proposed PUF, we found that it is very similar with (or identical to) the arbiter delay PUF. The authors claims that the proposed PUF is "configurable". But the configure factor K is actually the challenge of the PUF, which can be converted to an arbiter delay PUF model. So the author needs to show why the proposed PUF is different from arbiter delay PUF and the advantage over it. Also we know that the arbiter delay PUF is not resistant to machine learning attacks. So it is highly doubted whether the proposed PUF is machine learning resistant. The authors need to either prove it or show test result to verify it.
[1] Herder, Charles, et al. "Trapdoor computational fuzzy extractors and stateless cryptographically-secure physical unclonable functions." IEEE Transactions on Dependable and Secure Computing 14.1 (2016): 65-82.
[2] Wang, Ye, Xiaodan Xi, and Michael Orshansky. "Lattice PUF: A strong physical unclonable function provably secure against machine learning attacks." 2020 IEEE International Symposium on Hardware Oriented Security and Trust (HOST). IEEE, 2020.
Author Response
Response to Reviewer 1 Comments
Original Manuscript ID: Electronics-1421770
Original Article Title: “RPPUF: An Ultra-Lightweight Reconfigurable Pico-Physically Unclonable Function for Resource-constrained IoT Devices”
To: Electronics Editor
Re: Response to Reviewer
Dear Editor,
Thank you for allowing a resubmission of our manuscript, with an opportunity to address the reviewers’ comments. We appreciate editor and reviewers very much for their positive and constructive comments and suggestions on our manuscript entitled “RPPUF: An Ultra-Lightweight Reconfigurable Pico-Physically Unclonable Function for Resource-constrained IoT Devices”.
We are uploading (a) our point-by-point response to the comments (below) (response to reviewers), and (b) a clean updated manuscript without highlights (PDF main document).
We have studied reviewer’s comments carefully and have made a revision. We have tried our best to revise our manuscript according to the comments. Thank you and best regards.
Best regards,
Zhao Huang et al.
Corresponding Author:
Quan Wang
qwang@xidian.edu.cn
Reviewer:
The authors presented a design of reconfigurable Pico-PUF for resource-constrained IoT devices such as FPGAs. The authors claim to increase the CRP space from 1 (Pico-PUF) to 2^n. However, we found that the proposed design is not a novel one and does not show greatness than arbiter delay PUF. We also do not see any evidence or test result to show the machine learning resistance of the proposed PUF. Compared to existing strong PUF designs, the proposed PUF does not show advantages apart from reliability. In conclusion, I suggest rejecting this paper.
- The authors provide a detailed analysis of the reliability and uniformity, which looks reasonable. However, we are more interested in how to improve the reliability of the PUF with either hardware or software solution. And we also want to see analysis of the unpredictability to prove or explain why the proposed PUF has better resistance to machine learning attacks. It does not make sence to compare the machine learning resistance with 1-CRP Pico-PUF. There has already been works to show provable resistance to machine learning attacks [1][2]. So the authors need to either prove their high resistance or explain why the proposed PUF has advantage over these two PUFs.
- From the description of the proposed PUF, we found that it is very similar with (or identical to) the arbiter delay PUF. The authors claims that the proposed PUF is "configurable". But the configure factor K is actually the challenge of the PUF, which can be converted to an arbiter delay PUF model. So the author needs to show why the proposed PUF is different from arbiter delay PUF and the advantage over it. Also we know that the arbiter delay PUF is not resistant to machine learning attacks. So it is highly doubted whether the proposed PUF is machine learning resistant. The authors need to either prove it or show test result to verify it.
[1] Herder, Charles, et al. "Trapdoor computational fuzzy extractors and stateless cryptographically-secure physical unclonable functions." IEEE Transactions on Dependable and Secure Computing 14.1 (2016): 65-82.
[2] Wang, Ye, Xiaodan Xi, and Michael Orshansky. "Lattice PUF: A strong physical unclonable function provably secure against machine learning attacks." 2020 IEEE International Symposium on Hardware Oriented Security and Trust (HOST). IEEE, 2020.
To Reviewer:
The authors appreciate very much for your constructive comments on our manuscript, “RPPUF: An Ultra-Lightweight Reconfigurable Pico-Physically Unclonable Function for Resource-constrained IoT Devices” (Manuscript ID Electronics-1421770).
We have revised our manuscript according to your comments. The revision details are summarized as follows.
Point # 1: The authors provide a detailed analysis of the reliability and uniformity, which looks reasonable. However, we are more interested in how to improve the reliability of the PUF with either hardware or software solution. And we also want to see analysis of the unpredictability to prove or explain why the proposed PUF has better resistance to machine learning attacks. It does not make sence to compare the machine learning resistance with 1-CRP Pico-PUF. There has already been works to show provable resistance to machine learning attacks [1][2]. So the authors need to either prove their high resistance or explain why the proposed PUF has advantage over these two PUFs.
[1] Herder, Charles, et al. "Trapdoor computational fuzzy extractors and stateless cryptographically-secure physical unclonable functions." IEEE Transactions on Dependable and Secure Computing 14.1 (2016): 65-82.
[2] Wang, Ye, Xiaodan Xi, and Michael Orshansky. "Lattice PUF: A strong physical unclonable function provably secure against machine learning attacks." 2020 IEEE International Symposium on Hardware Oriented Security and Trust (HOST). IEEE, 2020.
Response 1: We are appreciative of the reviewer’s suggestion. The main purpose of our method is intended to enhance the hardware resource utilization of PUFs, that is, to study how to generate more PUF response bits per unit area while ensuring the reliability of that, rather than improving the resistance to Machine Learning-based Attacks. However, there are some improper descriptions in this paper which may cause misunderstanding for readers. Also, the novelty and contribution are not stated clearly. But, your comments provide us a new research direction in the future work.
Action 1: To carefully address your comment, we have revised and updated the manuscript as follows.
- Revise the Abstract,Introduction and Related work to highlight our novelty and motivation.
Abstract: (Page 1, Abstract)
“... Since IoT devices are typically resource-limited, conventional security solutions such as classical cryptography are no longer applicable. Physically unclonable function (PUF) is a hardware-based, low-cost alternative solution to provide...However, despite existing PUFs have exhibited good performance, they are not well suitable for effective application on resource-constrained IoT devices due to the limited number of challenge-response pairs (CRPs) generated per unit area and large hardware resource overhead... This confirms that the proposed RPPUF is ultra-lightweight with good performance, making it more appropriate and efficient to apply in FPAG-based IoT devices with constrained resources.”
Section 1 - Introduction: (Page 2-3, Section 1, Paragraph 3)
”...However, current studies on PUFs primarily focus on improving reliability and many PUFs (including PPUF) have been reported in the literature to undergo significant hardware overhead [29]. Therefore, they may be difficult to effectively implement on FPGA-based IoT devices [14], [24]..., especially for IoT devices with limited hardware resource. Our goal is to solve these problems to increase BPC ratio without compromising PUF performance. ”
Section 2 - Related Work: (Page 4, Section 2.2, Paragraph 2, Last Line 4)
”...This means that a new ultra-lightweight LR PUF architecture that fulfill the practical criteria of low-cost and more CRPs is both interesting and necessary. Therefore, in this article, we focus on the PPUF and how we can overcome these shortages by using reconfigurable logic. And, this is the novelty and motivation of our study. ”
- Revise the main contributions of our paper.(Page 3, Section 1, Paragraph 2)
“(1) A novel and ultra-lightweight RPPUF has been proposed.
(2) By embedding configurable logic structure, the hardware resource utilization of PUF can be improved significantly. To the best of our knowledge, this is the first study to focus on the practical application of PUF architectures.
(3) A new performance measure, i.e., BPC, has been introduced to evaluate the hardware resource utilization of PUFs.
(4) We conduct the experimental RPPUF hardware implemented on FPGAs. The results show that under the conditions of similar PUF performance and same CRP space size, the resource overhead is dramatically reduced, making it more practical in IoT applications. ”
- Add the discussion among the state of the artsand our method to highlight the efficiency of our RPPUF.
Section 4.6. - Comparison with State of the Art. (Page 14, Section 4.6)
“In this section, a performance comparison of the proposed RPPUF structure against the state-of-the-arts has been performed...” (Page 14, Section 4.6, Paragraph 1)
“From Table 5, it can be seen that these PUF architectures except lattice PUF are delaybased design. The performance results for all PUFs are ...” (Page 14, Section 4.6, Paragraph 2)
- Delete the inappropriate statements from original Section 3.3 - Security Analysis.
- Add the limitation to ML-based attack to current Section 3.3 - Characteristics of the RPPUF.
“5) However, as PPUF changes from weak to “strong”, it is itself vulnerable to security threats such as machine learning (ML)-based attacks accordingly; ” (Page 6, current Section 3.3, Point 5)
- Add this point to the future work part in Section 5.
“...Also, the RPPUF may be not resistant to ML-based attacks since they can be formulated as a linear additive delay model. Therefore, future work will include decreasing the data dependency of PUF responses and enhancing the security to against ML-based modelling attacks.” (Page 15, Section 5, Paragraph 2, Line 4)
Point # 2: From the description of the proposed PUF, we found that it is very similar with (or identical to) the arbiter delay PUF. The authors claims that the proposed PUF is "configurable". But the configure factor K is actually the challenge of the PUF, which can be converted to an arbiter delay PUF model. So the author needs to show why the proposed PUF is different from arbiter delay PUF and the advantage over it. Also we know that the arbiter delay PUF is not resistant to machine learning attacks. So it is highly doubted whether the proposed PUF is machine learning resistant. The authors need to either prove it or show test result to verify it.
Response 2: We are appreciative of the reviewer’s suggestion. However, the authors think that it is different from arbiter PUF. 1) The delay elements that utilized are different. Arbiter PUF generates a 1 -bit PUF response by comparing the wire/line delay between two cross or parallel multiplexer chains. However, our method primary uses the delay between two configurable inverter (INV) chains to determine an output. The configurable inverter (INV) chain is also called configurable ring oscillator technique which is proposed by Maiti et al. in 2011 [1], [2]. In our paper, we exploit the configurable RO but remove the feedback line to enhance reliability and increase key space of Pico PUF. 2) The sources are different. As stated in the paper, our solution is proposed based on the Pico PUF structure of Gu et al. [3] and also referring to the reliability-improved butterfly PUF of Xu et al. [4]. Actually, they are all different from arbiter PUF. 3) The configurable targets are different. The challenges for arbiter PUF are intended to configure each switch component to obtain different transition races [5]. For example, if input = 0, the paths go straight through, while for input = 1 they are crossed. However, our method doesn’t require this operation. We only utilize a multiplexer to select different INVs to compete for PUF response bit generation. Despite these PUFs all utilize the delay differences as a PUF response source, their structure and operating principles are different.
Moreover, our approach is essentially to increase the hardware resource utilization of PUF, which means using the least hardware resources to generate as much response output bits as possible. This is the goal of this paper, but not to improve resistance to machine learning attacks. At this point, however, due to our careless, we are very sorry that the main novelty of our paper may not describe very clearly.
[1] Maiti, A.; Schaumont, P. Improved Ring Oscillator PUF: An FPGA-friendly Secure Primitive. Journal of Cryptology, vol. 24, pp. 375-397, April 2011.
[2] Huang, Z.; and Wang, Q. A PUF-based Unified Identity Verification Framework for Secure IoT Hardware via Device Authentication. World Wide Web-Internet and Web Information Systems, vol. 23, no. 2, pp. 1057–1088, March 2020.
[3] Gu, C.Y.; Hanley, N.; and O’Neill, M. Improve Reliability of FPGA-Based PUF Identification Generator Design. ACM Trans. on Reconfigurable Technology and Systems, vol. 10, no. 3, pp. 20:1–20:23, May 2017.
[4] Xu, X.M.; Liang, H.G.; Huang, Z.F.; Jiang, C.Y.; Ouyang, Y.M.; Fang, X.S.; Ni, T.M.; and Yi, M.X. A Highly Reliable Butterfly PUF in SRAM-based FPGA. IEICE Electronics Express, vol. 14, no. 14, pp. 1–6, July 2017.
[5] Lim, D.; Lee, J.W.; Gassend, B., Suh, E. Extracting Secret Keys From Integrated Circuits. IEEE Trans. on VLSI System, vol. 13, no. 10, pp. 1200-1205, October 2005.
Action 2: To carefully address your comment, we have updated the manuscript and conducted the following revisions.
- Delete the inappropriate statements from original Section 3.3 - Security Analysis.
- Add the limitation to ML-based attack to current Section 3.3 - Characteristics of the RPPUF.
“5) However, as PPUF changes from weak to “strong”, it is itself vulnerable to security threats such as machine learning (ML)-based attacks accordingly; ” (Page 6, current Section 3.3, Point 5)
- Add this point to the future work part in Section 5.
“...Also, the RPPUF may be not resistant to ML-based attacks since they can be formulated as a linear additive delay model. Therefore, future work will include decreasing the data dependency of PUF responses and enhancing the security to against ML-based modelling attacks.” (Page 15, Section 5, Paragraph 2, Line 4)
- Revise the Abstract, Introduction and Related work to highlight our novelty and motivation.
Abstract: (Page 1, Abstract)
“... Since IoT devices are typically resource-limited, conventional security solutions such as classical cryptography are no longer applicable. Physically unclonable function (PUF) is a hardware-based, low-cost alternative solution to provide...However, despite existing PUFs have exhibited good performance, they are not well suitable for effective application on resource-constrained IoT devices due to the limited number of challenge-response pairs (CRPs) generated per unit area and large hardware resource overhead... This confirms that the proposed RPPUF is ultra-lightweight with good performance, making it more appropriate and efficient to apply in FPAG-based IoT devices with constrained resources.”
Section 1 - Introduction: (Page 2-3, Section 1, Paragraph 3)
”...However, current studies on PUFs primarily focus on improving reliability and many PUFs (including PPUF) have been reported in the literature to undergo significant hardware overhead [29]. Therefore, they may be difficult to effectively implement on FPGA-based IoT devices [14], [24]..., especially for IoT devices with limited hardware resource. Our goal is to solve these problems to increase BPC ratio without compromising PUF performance. ”
Section 2 - Related Work: (Page 4, Section 2.2, Paragraph 2, Last Line 4)
”...This means that a new ultra-lightweight LR PUF architecture that fulfill the practical criteria of low-cost and more CRPs is both interesting and necessary. Therefore, in this article, we focus on the PPUF and how we can overcome these shortages by using reconfigurable logic. And, this is the novelty and motivation of our study. ”

Reviewer 2 Report
The proposed PUF construct embeds an arbiter PUF into an existing weak PUF construct deployed in an FPGA, called the Pico-PUF [1].
Hence, when claiming that the design technique used turns a weak PPUF into a "strong" PUF, one cannot ignore that this is mostly
due to the incorporation of this widely known and thoroughly studied strong PUF primitive.
The novelty of this work needs to be clarified, since the PUF structure proposed here is very similiar to the one proposed in [2].
Furthermore, a novel compact PUF with improved area efficiency and reliabiliy, as well as well-supported uniqueness results (using NIST tests) is already published in [3].
These works must be included in the discussion of results, and it should be made clear whether there is a significant improvement on existing compact PUF designs.
[1] Gu, C.; Chang, C.H.; Liu, W.; Hanley, N.; Miskelly, J.; O’Neill, M. A large-scale comprehensive evaluation of single-slice ring
oscillator and PicoPUF bit cells on 28-nm Xilinx FPGAs. J. Cryptogr. Eng. 2020, 11, 1–12.
[2] C. Gu, W. Liu, Y. Cui, N. Hanley, M. O'Neill and F. Lombardi, "A Flip-Flop Based Arbiter Physical Unclonable Function (APUF) Design with High Entropy and Uniqueness for FPGA Implementation," in IEEE Transactions on Emerging Topics in Computing, doi: 10.1109/TETC.2019.2935465.
[3] Della Sala, R., Bellizia, D. and Scotti, G., 2021. A novel ultra-compact fpga puf: The dd-puf. Cryptography, 5(3), p.23.
There are many sentences with an unclear meaning, some missing words and unfinished sentences (e.g.: "Also,." in page 3).
Some examples are:
English language errors/typos:
page 1 : "an low-cost" -> "a low-cost"
page 2 : "an PUF" -> "a PUF"
page 3 : "an latch" -> "a latch" , "an 1-bit" -> "a 1-bit", missing "the" in "Another typical delay-based PFU is (the) RO PUF".
page 6 : "pbulicly" , "liner" -> linear
page 6 : "This makes the RPPUF circuit has no fixed.."
page 7 : "an challenge" -> "a challenge "
Not recommended writing style:
page 3: "so we call them delay-based PUFs.."
page 5: "In implementation on FPGAs.."
page 6: "such attacks are very potent implementation on FPGAs.."
Unclear meaning:
page 3 : " Arbiter PUFs generate a 1-bit PUF response by the delay difference ..."
The autors probably mean: "... generate a 1 -bit PUF response by comparing the delay between two parallel multiplexer chains"
page 6: "zhang et al. has proven that MLs can successfuly.." - what does MLs mean ? Machine Learnings?
page 7: "From Fig.5, we can conduct that .." should it be "conclude?"
Author Response
Response to Reviewer 2 Comments
Original Manuscript ID: Electronics-1421770
Original Article Title: “RPPUF: An Ultra-Lightweight Reconfigurable Pico-Physically Unclonable Function for Resource-constrained IoT Devices”
To: Electronics Editor
Re: Response to Reviewer
Dear Editor,
Thank you for allowing a resubmission of our manuscript, with an opportunity to address the reviewers’ comments. We appreciate editor and reviewers very much for their positive and constructive comments and suggestions on our manuscript entitled “RPPUF: An Ultra-Lightweight Reconfigurable Pico-Physically Unclonable Function for Resource-constrained IoT Devices”.
We are uploading (a) our point-by-point response to the comments (below) (response to reviewers), and (b) a clean updated manuscript without highlights (PDF main document).
We have studied reviewer’s comments carefully and have made a revision. We have tried our best to revise our manuscript according to the comments. Thank you and best regards.
Best regards,
Zhao Huang et al.
Corresponding Author:
Quan Wang
qwang@xidian.edu.cn
Reviewer:
The proposed PUF construct embeds an arbiter PUF into an existing weak PUF construct deployed in an FPGA, called the Pico-PUF [1]. Hence, when claiming that the design technique used turns a weak PPUF into a "strong" PUF, one cannot ignore that this is mostly due to the incorporation of this widely known and thoroughly studied strong PUF primitive. The novelty of this work needs to be clarified, since the PUF structure proposed here is very similiar to the one proposed in [2]. Furthermore, a novel compact PUF with improved area efficiency and reliabiliy, as well as well-supported uniqueness results (using NIST tests) is already published in [3]. These works must be included in the discussion of results, and it should be made clear whether there is a significant improvement on existing compact PUF designs.
[1] Gu, C.; Chang, C.H.; Liu, W.; Hanley, N.; Miskelly, J.; O’Neill, M. A large-scale comprehensive evaluation of single-slice ring oscillator and PicoPUF bit cells on 28-nm Xilinx FPGAs. J. Cryptogr. Eng. 2020, 11, 1–12.
[2] C. Gu, W. Liu, Y. Cui, N. Hanley, M. O'Neill and F. Lombardi, "A Flip-Flop Based Arbiter Physical Unclonable Function (APUF) Design with High Entropy and Uniqueness for FPGA Implementation," in IEEE Transactions on Emerging Topics in Computing, doi: 10.1109/TETC.2019.2935465.
[3] Della Sala, R., Bellizia, D. and Scotti, G., 2021. A novel ultra-compact fpga puf: The dd-puf. Cryptography, 5(3), p.23.
There are many sentences with an unclear meaning, some missing words and unfinished sentences (e.g.: "Also,." in page 3).
Some examples are:
English language errors/typos:
page 1: "an low-cost" -> "a low-cost"
page 2: "an PUF" -> "a PUF"
page 3: "an latch" -> "a latch" , "an 1-bit" -> "a 1-bit", missing "the" in "Another typical delay-based PFU is (the) RO PUF".
page 6: "pbulicly" , "liner" -> linear
page 6: "This makes the RPPUF circuit has no fixed.."
page 7: "an challenge" -> "a challenge "
Not recommended writing style:
page 3: "so we call them delay-based PUFs.."
page 5: "In implementation on FPGAs.."
page 6: "such attacks are very potent implementation on FPGAs.."
Unclear meaning:
page 3: " Arbiter PUFs generate a 1-bit PUF response by the delay difference ..." The autors probably mean: "... generate a 1 -bit PUF response by comparing the delay between two parallel multiplexer chains"
page 6: "zhang et al. has proven that MLs can successfuly.." - what does MLs mean ? Machine Learnings?
page 7: "From Fig.5, we can conduct that .." should it be "conclude?"
To Reviewer:
The authors appreciate very much for your constructive comments on our manuscript, “RPPUF: An Ultra-Lightweight Reconfigurable Pico-Physically Unclonable Function for Resource-constrained IoT Devices” (Manuscript ID Electronics-1421770).
We have revised our manuscript according to your comments. The revision details are summarized as follows.
Point # 1: The novelty of this work needs to be clarified, since the PUF structure proposed here is very similiar to the one proposed in [2].
[1] Gu, C.; Chang, C.H.; Liu, W.; Hanley, N.; Miskelly, J.; O’Neill, M. A large-scale comprehensive evaluation of single-slice ring oscillator and PicoPUF bit cells on 28-nm Xilinx FPGAs. J. Cryptogr. Eng. 2020, 11, 1–12.
[2] C. Gu, W. Liu, Y. Cui, N. Hanley, M. O'Neill and F. Lombardi, "A Flip-Flop Based Arbiter Physical Unclonable Function (APUF) Design with High Entropy and Uniqueness for FPGA Implementation," in IEEE Transactions on Emerging Topics in Computing, doi: 10.1109/TETC.2019.2935465.
[3] Della Sala, R., Bellizia, D. and Scotti, G., 2021. A novel ultra-compact fpga puf: The dd-puf. Cryptography, 5(3), p.23.
Response 1: We are very sorry for our negligence of inappropriate description about the novelty, motivation and contribution. Since the main goal of our method is intended to enhance hardware resource utilization of PUFs, which means using the least hardware resources to generate as much response output bits as possible, the novelty and contribution are not stated clearly.
Action 1: To carefully address your comment, we have revised and updated the manuscript as follows.
- Revise the Abstract,Introduction and Related work to highlight our novelty and motivation.
Abstract: (Page 1, Abstract)
“... Since IoT devices are typically resource-limited, conventional security solutions such as classical cryptography are no longer applicable. Physically unclonable function (PUF) is a hardware-based, low-cost alternative solution to provide...However, despite existing PUFs have exhibited good performance, they are not well suitable for effective application on resource-constrained IoT devices due to the limited number of challenge-response pairs (CRPs) generated per unit area and large hardware resource overhead... This confirms that the proposed RPPUF is ultra-lightweight with good performance, making it more appropriate and efficient to apply in FPAG-based IoT devices with constrained resources.”
Section 1 - Introduction: (Page 2-3, Section 1, Paragraph 3)
”...However, current studies on PUFs primarily focus on improving reliability and many PUFs (including PPUF) have been reported in the literature to undergo significant hardware overhead [29]. Therefore, they may be difficult to effectively implement on FPGA-based IoT devices [14], [24]..., especially for IoT devices with limited hardware resource. Our goal is to solve these problems to increase BPC ratio without compromising PUF performance. ”
Section 2 - Related Work: (Page 4, Section 2.2, Paragraph 2, Last Line 4)
”...This means that a new ultra-lightweight LR PUF architecture that fulfill the practical criteria of low-cost and more CRPs is both interesting and necessary. Therefore, in this article, we focus on the PPUF and how we can overcome these shortages by using reconfigurable logic. And, this is the novelty and motivation of our study. ”
- Revise the main contributions of our paper.(Page 3, Section 1, Paragraph 2)
“(1) A novel and ultra-lightweight RPPUF has been proposed.
(2) By embedding configurable logic structure, the hardware resource utilization of PUF can be improved significantly. To the best of our knowledge, this is the first study to focus on the practical application of PUF architectures.
(3) A new performance measure, i.e., BPC, has been introduced to evaluate the hardware resource utilization of PUFs.
(4) We conduct the experimental RPPUF hardware implemented on FPGAs. The results show that under the conditions of similar PUF performance and same CRP space size, the resource overhead is dramatically reduced, making it more practical in IoT applications. ”
Point # 2: Furthermore, a novel compact PUF with improved area efficiency and reliabiliy, as well as well-supported uniqueness results (using NIST tests) is already published in [3]. These works must be included in the discussion of results, and it should be made clear whether there is a significant improvement on existing compact PUF designs.
Response 2: We are appreciative of the reviewer’s suggestion. Just like what the reviewer suggested, the discussion of the paper [3] will help to improve our paper and make readers more understand our work.
Action 2: We updated the manuscript by ….
- Add a new Section. (Page 14, New Section 4.6)
Section 4.6 Comparison with State of the Art
- The suggested papers are cited in this paper. (Page 17, REFERENCE, Literature [35] - [37])
[35]. Gu, C.Y.; Chang, C.H.; Liu, W.Q.; Hanley, N.; Miskelly, J.; and O’Neill, M. A Large-scale Comprehensive Evaluation of Single-slice Ring Oscillator and PicoPUF Bit Cells on 28-nm Xilinx FPGAs. Journal of Cryptographic Engineering, vol. 11, no. 3, pp. 227–238, December 2020.
[36]. Gu, C.Y.; Liu, W.Q.; Cui, Y.J.; Hanley, N.; O’Neill, M.; and Lombardi, F. A Flip-Flop Based Arbiter Physical Unclonable Function (APUF) Design with High Entropy and Uniqueness for FPGA Implementation. IEEE Trans. on Emerging Topics in Computing, pp. 1–13, September 2019. DOI: 10.1109/TETC.2019.2935465.
[37]. Sala, R.D.; Bellizia, D.; Scotti, G. A Novel Ultra-Compact FPGA PUF: The DD-PUF. Cryptography, vol. 23, no. 5, pp. 1–18, September 2021.
- Add the discussion among the state of the artsand our method to highlight the efficiency of our RPPUF.
Section 4.6. - Comparison with State of the Art. (Page 14, Section 4.6)
“In this section, a performance comparison of the proposed RPPUF structure against the state-of-the-arts has been performed...” (Page 14, Section 4.6, Paragraph 1)
“From Table 5, it can be seen that these PUF architectures except lattice PUF are delaybased design. The performance results for all PUFs are ...” (Page 14, Section 4.6, Paragraph 2)
Point # 3: There are many sentences with an unclear meaning, some missing words and unfinished sentences (e.g.: "Also,." in page 3).
Some examples are:
English language errors/typos:
page 1: "an low-cost" -> "a low-cost"
page 2: "an PUF" -> "a PUF"
page 3: "an latch" -> "a latch" , "an 1-bit" -> "a 1-bit", missing "the" in "Another typical delay-based PFU is (the) RO PUF".
page 6: "pbulicly", "liner" -> linear
page 6: "This makes the RPPUF circuit has no fixed.."
page 7: "an challenge" -> "a challenge "
Not recommended writing style:
page 3: "so we call them delay-based PUFs.."
page 5: "In implementation on FPGAs.."
page 6: "such attacks are very potent implementation on FPGAs.."
Unclear meaning:
page 3: " Arbiter PUFs generate a 1-bit PUF response by the delay difference ..." The autors probably mean: "... generate a 1 -bit PUF response by comparing the delay between two parallel multiplexer chains"
page 6: "zhang et al. has proven that MLs can successfuly.." - what does MLs mean ? Machine Learnings?
page 7: "From Fig.5, we can conduct that .." should it be "conclude?"
Response 3: Thank you very much for your carefully checking. We have carefully checked and revised the spelling mistakes by ourselves and also invited a native English speaking individual to help us check and polish up our paper.
Action 3: To carefully address your comment, we have revised the spelling/grammatical mistakes and typos. For example:
- Correct the English language errors/typos:.
Current Page 2, Section 1, Paragraph 2, Line 1: "...(PUF) is a low-cost,... "
Current Page 2, Section 1, Paragraph 2, Last Line 8: "..., that is, a PUF instance ..."
Current Page 3, Section 2.1, Paragraph 1, Last Line 8: "...which uses a latch structure..." , Paragraph 2, Last Line 5: "...generate a 1-bit PUF...", Last Line 1: Add "the" in "Another typical delay-based PUF is the RO PUF ".
page 6: "pbulicly", "liner" -> linear (Since we delete the original Section 3.3, thus these parts don't exist anymore)
page 6: "This makes the RPPUF circuit has no fixed.." (Since we delete the original Section 3.3, thus these parts don't exist anymore)
Current Page 7, Section 4.1, Paragraph 1, Line 2: "...and a challenge ..."
- Modify the Not recommended writing sentences:
Current Page 2, Section 1, Paragraph 2, Last Line 4: "..., they are named delay-based PUFs .."
Current Page 6, Section 3.2, Paragraph 2, Line 1: "When implemented on FPGAs, .."
page 6: "such attacks are very potent implementation on FPGAs.."
(Delete the original Section 3.3, thus these parts don't exist anymore)
- Revise the sentences with unclear meaning:
Current Page 3, Section 2.1, Paragraph 2, Line 5: "Arbiter PUFs generate a 1-bit PUF response by comparing the delay difference between parallel multiplexer chains [25]. "
page 6: "zhang et al. has proven that MLs can successfuly.." - what does MLs mean ? Machine Learnings? (Since we delete the original Section 3.3, thus these parts don't exist anymore)
Current Page 7, Section 4.1, Last Paragraph, Last Line 2: "From Fig. 6, we can conclude that .."
- Other mistakes:
Current Page 8, Section 4.2, Last Paragraph, Last Line: "the quantitative comparision..." to “"the quantitative comparison...”
Current Page 8, Section 4.2, Paragraph 3, Line 2: "...an 128-bit ..." to "...a 128-bit ..."
Current Page 1, Abstract, Last Line 2: "...in FPAG-based IoT..." to "...in FPGA-based IoT..."
Current Page , Section 1, Paragraph 2, Last Line 2: "... authentication procols ..." to "... authentication protocols ..."
...
For more details, please kindly refer to our revised manuscript, thank you.

Reviewer 3 Report
The paper presents a study regarding the design of a PUF for resource constrained IoT devices in order to enhance their hardware security.
The paper is well organized and written.
Several concerns (and suggestions to improve) about this manuscript, that this Reviewer has identified are listed bellow:
- The Abstract should be adjusted, slightly in order to point out the "solution" described herein in this manuscript. For example, if the main target is IoT devices with limited computing capacity, this should be highlighted in order to bring out more interest for the readers.
- In the introduction section, the problem of IoT security is not well presented and described/classified. Therefore, I would suggest to the Authors to improve this Introduction Section by enhancing the state-of-the-art regarding the IoT Security paradigm.
- For example, a recently published paper:
https://ieeexplore.ieee.org/abstract/document/8792895 that the
Authors should cite, is a starting point to improve the IoT security state-of-the art.
Best Regards,
The Reviewer
Author Response
Response to Reviewer 3 Comments
Original Manuscript ID: Electronics-1421770
Original Article Title: “RPPUF: An Ultra-Lightweight Reconfigurable Pico-Physically Unclonable Function for Resource-constrained IoT Devices”
To: Electronics Editor
Re: Response to Reviewer
Dear Editor,
Thank you for allowing a resubmission of our manuscript, with an opportunity to address the reviewers’ comments. We appreciate editor and reviewers very much for their positive and constructive comments and suggestions on our manuscript entitled “RPPUF: An Ultra-Lightweight Reconfigurable Pico-Physically Unclonable Function for Resource-constrained IoT Devices”.
We are uploading (a) our point-by-point response to the comments (below) (response to reviewers), and (b) a clean updated manuscript without highlights (PDF main document).
We have studied reviewer’s comments carefully and have made a revision. We have tried our best to revise our manuscript according to the comments. Thank you and best regards.
Best regards,
Zhao Huang et al.
Corresponding Author:
Quan Wang
qwang@xidian.edu.cn
Reviewer:
The paper presents a study regarding the design of a PUF for resource constrained IoT devices in order to enhance their hardware security.
The paper is well organized and written.
Several concerns (and suggestions to improve) about this manuscript, that this Reviewer has identified are listed bellow:
- The Abstract should be adjusted, slightly in order to point out the "solution" described herein in this manuscript. For example, if the main target is IoT devices with limited computing capacity, this should be highlighted in order to bring out more interest for the readers.
- In the introduction section, the problem of IoT security is not well presented and described/classified. Therefore, I would suggest to the Authors to improve this Introduction Section by enhancing the state-of-the-art regarding the IoT Security paradigm.For example, a recently published paper: https://ieeexplore.ieee.org/abstract/document/8792895 that the Authors should cite, is a starting point to improve the IoT security state-of-the art.
Best Regards,
The Reviewer
To Reviewer:
The authors appreciate very much for your constructive comments on our manuscript, “RPPUF: An Ultra-Lightweight Reconfigurable Pico-Physically Unclonable Function for Resource-constrained IoT Devices” (Manuscript ID Electronics-1421770).
We have revised our manuscript according to your comments. The revision details are summarized as follows.
Point # 1: The Abstract should be adjusted, slightly in order to point out the "solution" described herein in this manuscript. For example, if the main target is IoT devices with limited computing capacity, this should be highlighted in order to bring out more interest for the readers.
Response 1: We are appreciative of the reviewer’s suggestion. Just like what the reviewer suggested, the detail and highlight of the main target will interest the readers.
Action 1: To carefully address you comment, we have revised and updated the manuscript as follows.
- Revise the Abstract and Introduction to highlight the main target of our solution is to secure resource-constrained IoT devices.
Abstract: (Page 1, Abstract)
“... Since IoT devices are typically resource-limited, conventional security solutions such as classical cryptography are no longer applicable. Physically unclonable function (PUF) is a hardware-based, low-cost alternative solution to provide...However, despite existing PUFs have exhibited good performance, they are not well suitable for effective application on resource-constrained IoT devices due to the limited number of challenge-response pairs (CRPs) generated per unit area and large hardware resource overhead... This confirms that the proposed RPPUF is ultra-lightweight with good performance, making it more appropriate and efficient to apply in FPAG-based IoT devices with constrained resources.”
Section 1 - Introduction: (Page 1-2, Section 1)
“...However, these IoT devices usually have limited hardware resources and poor self-protection capabilities, while classical encryption solutions are often resource-/time- intensive [7], [8], [14], [29], making them unavailable for IoT applications. In view of this, it is both necessary and urgent to develop lightweight solutions to ensure the security of IoT devices.” (Page 2, Section 1, Paragraph 1, Line 4)
”...However, current studies on PUFs primarily focus on improving reliability and many PUFs (including PPUF) have been reported in the literature to undergo significant hardware overhead [29]. Therefore, they may be difficult to effectively implement on FPGA-based IoT devices [14], [24]..., especially for IoT devices with limited hardware resource. Our goal is to solve these problems to increase BPC ratio without compromising PUF performance. ” (Page 2, Section 1, Last Paragraph 1, Last Line 6)
Comment # 2: In the introduction section, the problem of IoT security is not well presented and described/classified. Therefore, I would suggest to the Authors to improve this Introduction Section by enhancing the state-of-the-art regarding the IoT Security paradigm. For example, a recently published paper: https://ieeexplore.ieee.org/abstract/document/8792895 that the Authors should cite, is a starting point to improve the IoT security state-of-the art.
Response 2: We are very sorry for our negligence of inappropriate description about the IoT security problem.
Action 2: To carefully address you comment, we have revised and updated the manuscript as follows.
- The suggested Paper is cited in reference. (Page 15, REFERENCE, Literature 4)
[4]. Tudosa, I.; Picariello, F.; Balestrieri, E.; Vito, L.D.; Lamonaca, F. Hardware Security in IoT era: the Role of Measurements and Instrumentation. in Proc. of 2019 II Workshop on Metrology for Industry 4.0 and IoT (MetroInd4.0&IoT), Naples, Italy, 4-6 June 2019; pp. 285–290.
- Add a Threats model Figure.
(Page 2, Section 1, Figure 1)
- Add corresponding description about the problemof IoT security in our revised version.
“...Fig. 1 presents the security threats faced by each layer of the IoT framework [4], [5]. From the perspective of IoT framework, it can be divided into four layers... ” (Page 1, Section 1, Paragraph 1, Last Line 7)

Round 2
Reviewer 1 Report
Thanks for the authors to provide detailed explanation for my questions and modifying the manuscript. With the answers and the edits, I think the paper is much better, while still there are some points I think can be improved.
1. It' s good that the authors added comparison results with existing state-of-the-art PUF works. I think the comparison can be still enhanced by adding comparison of CRP#, PUF type, etc.
2. I agree that the intent of this work is not to improve machine learning resistance of PUF, but we do want to see comparison of machine learning resistance with existing state-of-the-art works and good explanation and analysis for the comparison. I think the authors can improve the work with this information.
3. I thank the authors for providing explanation of differentiating the proposed PUF with arbiter delay PUFBut my point is that we want to see analysis evidence or experimental results to show the machine learning resistance of the proposed PUF. I know this may not be the key novelty of this work, but for a strong PUF to be really applied in security applications, we expect the PUF to be machine learning resistant as a baseline.
Author Response
Response to Reviewer 1 Comments
Original Manuscript ID: Electronics-1421770
Original Article Title: “RPPUF: An Ultra-Lightweight Reconfigurable Pico-Physically Unclonable Function for Resource-constrained IoT Devices”
To: Electronics Editor
Re: Response to Reviewer
Dear Editor,
Thank you for allowing a re-submission of our manuscript, with an opportunity to address the reviewers’ comments. We appreciate editor and reviewers very much for their positive and constructive comments and suggestions on our manuscript entitled “RPPUF: An Ultra-Lightweight Reconfigurable Pico-Physically Unclonable Function for Resource-constrained IoT Devices”.
We are uploading (a) our point-by-point response to the comments (below) (response to reviewers), (b) a updated manuscript with modification highlights (PDF main document), and (c) a clean updated manuscript without highlights (PDF main document).
We have studied reviewer’s comments carefully and have made a revision. We have tried our best to revise our manuscript according to the comments. Thank you and best regards.
Best regards,
Zhao Huang et al.
Corresponding Author:
Quan Wang
qwang@xidian.edu.cn
Reviewer:
Thanks for the authors to provide detailed explanation for my questions and modifying the manuscript. With the answers and the edits, I think the paper is much better, while still there are some points I think can be improved.
- It' s good that the authors added comparison results with existing state-of-the-art PUF works. I think the comparison can be still enhanced by adding comparison of CRP#, PUF type, etc.
- I agree that the intent of this work is not to improve machine learning resistance of PUF, but we do want to see comparison of machine learning resistance with existing state-of-the-art works and good explanation and analysis for the comparison. I think the authors can improve the work with this information.
- I thank the authors for providing explanation of differentiating the proposed PUF with arbiter delay PUF. But my point is that we want to see analysis evidence or experimental results to show the machine learning resistance of the proposed PUF. I know this may not be the key novelty of this work, but for a strong PUF to be really applied in security applications, we expect the PUF to be machine learning resistant as a baseline.
To Reviewer:
The authors appreciate very much for your constructive comments on our manuscript, “RPPUF: An Ultra-Lightweight Reconfigurable Pico-Physically Unclonable Function for Resource-constrained IoT Devices” (Manuscript ID Electronics-1421770).
We have revised our manuscript according to your comments. The revision details are summarized as follows.
Point # 1: It' s good that the authors added comparison results with existing state-of-the-art PUF works. I think the comparison can be still enhanced by adding comparison of CRP#, PUF type, etc.
Response 1: We are appreciative of the reviewer’s suggestion. Just like what the reviewer suggested, the adding of CRP#, PUF type, etc. will help readers better understand the comparison of various PUFs.
Action 1: To carefully address your comment, we have revised and updated the manuscript as follows.
- Add the PUF metric of PUF type to original Table 5.(see Page 15, current Section 4.7, current Table 7, Column 3 )
- Add the PUF metric of CRP# to new Table 5,6 and 9.
(see Page 15, new Section 4.6, new Table 5, Column 5) -- (see Page 15, new Section 4.6, new Table 6, Column 2 )
(see Page 17, current Section 4.7, new Table 9, Column 3 )
- Add newPUF metric of NCBPR to original Table 6. (see Page 16, current Section 4.7, current Table 8, Column 2)
- Provide corresponding explanation about them.
“...Since PPUF, 4DBPUF and DD-PUF are weak PUFs (see Table 7), their NCBPR value is 1. As can be seen from Table 8 that among strong PUFs, RPPUF has ...” (see Page 16, current Section 4.7, Paragraph 2, Line 9 - 10)
“...Since ML attacks are ineffective against weak PUFs, Table 9 only provides the results of strong PUFs ...” (see Page 16, current Section 4.7, Paragraph 3, Line 2 - 3)
“...In addition, we also summary the number of input challenge bits required to produce 1-bit of PUF response, which is abbreviated as NCBPR. ...” (see Page 16, current Section 4.7, Paragraph 2, Line 9 - 11)
“...using Open source Python packages and learned with training CRP sample sets varying in size from 20,000 to 100,000. ...” (see Page 14, new Section 4.6, Paragraph 2, Line 5 - 6)
...
For more details, please kindly refer to our revised manuscript, thank you.
Point # 2: I agree that the intent of this work is not to improve machine learning resistance of PUF, but we do want to see comparison of machine learning resistance with existing state-of-the-art works and good explanation and analysis for the comparison. I think the authors can improve the work with this information.
Response 2: We are appreciative of the reviewer’s suggestion. Since the ML-resistant research of strong PUFs are current research topics and key issues, a comparison of machine learning resistance with existing state-of-the-art works and good explanation of that are necessary and interesting for readers.
Action 2: To carefully solve your comment, we have updated the manuscript and conducted the following revisions.
- Add new Section of ML Attacks Results. (see Page 14, new Section 4.6, new Title 4.6)
4.6. ML Attacks Results
- Add new Figure 13 to show the Attack Results. (see Page 14, new Section 4.6, new Figure 13)
- Add new Table 5 and 6 to illustrate the Prediction Rates and Training Time for each ML attack. (see Page 15, new Section 4.6, new Table 5 - 6)
(see Page 15, new Section 4.6, new Table 5)
(see Page 15, new Section 4.6, new Table 6)
- Providing Corresponding Descriptions about the Attack Results of our RPPUF. (see Page 14 - 15, new Section 4.6)
“Ideally, PUFs are unpredictable, unclonable, and tamper evident ... we will evaluate the capability of the proposed RPPUF to resist ML attacks.” (see Page 14, new Section 4.6, Paragraph 1)
-------Description for new Figure 13.
“To evaluate the ML-resistance of the proposed RPPUF architecture, we utilize ... using these ML attack models and the prediction rates are shown in Fig. 13.” (see Page 14, new Section 4.6, Paragraph 2)
“In Fig. 13, it is obviously that RPPUF exhibits a little better resistance to ... thereby enabling them to achieve high accuracy. .” (see Page 14, new Section 4.6, Paragraph 3)
-------Description for new Table 5 and 6.
“The final prediction rates, including ... the attack success efficiency for DT is the highest.” (see Page 14 - 15, new Section 4.6, Paragraph 4)
“In general, the ML-resistance of the proposed RPPUF is ... it needs to be further strengthened.” (see Page 15, new Section 4.6, Paragraph 2)
- Add a new Table fora comparison of machine learning resistance with existing state-of-the-art works in original Fig. 9. (see Page 17, current Section 4.7, new Table 9)
- Provide corresponding descriptionof new Table 9, including the comparison of ML-resistance for our RPPUF and existing state-of-the-art Works.
“Also, the comparison of security analysis for different PUF designs is provided in Table 9 ... but it is worse than FF-APUF and lattice PUF design.” (see Page 16 - 17, current Section 4.7, Last Paragraph)
Point # 3: I thank the authors for providing explanation of differentiating the proposed PUF with arbiter delay PUF. But my point is that we want to see analysis evidence or experimental results to show the machine learning resistance of the proposed PUF. I know this may not be the key novelty of this work, but for a strong PUF to be really applied in security applications, we expect the PUF to be machine learning resistant as a baseline.
Response 3: We are very sorry for our negligence of descriptions about the ML attack results for RPPUF. Since ML-resistance is an important performance metric for PUF, this part is missing in our manuscript. Just like the reviewer suggested, the ML attack results of different ML techniques for the RPPUF design will help readers to better understand the security of our scheme. In this version, we launch the modeling attacks on our RPPUF using 5 traditional ML models, that is, DT, RF, SVM, ANN, LR with the size of training CRP sample sets varying from 20K to 100K (increased by 20K each time).
Action 3: To carefully address your comment, we have revised and updated the manuscript as follows.
- Add the results for Machine Learning attacks such as SVM, ANN, LR, DT, RF.
- Add new Figures to show the Security results
- Provide corresponding explanation about new Figures.
- Add new Section of ML Attacks Results. (see Page 14, new Section 4.6, new Title 4.6)
4.6. ML Attacks Results
- Add new Figure 13 to show the Attack Results. (see Page 14, new Section 4.6, new Figure 13)
- Add new Table 5 and 6 to illustrate the Prediction Rates and Training Time for each ML attack. (see Page 15, new Section 4.6, new Table 5 - 6)
(see Page 15, new Section 4.6, new Table 5)
(see Page 15, new Section 4.6, new Table 6)
- Providing Corresponding Descriptions about the Attack Results of our RPPUF. (see Page 14 - 15, new Section 4.6)
“Ideally, PUFs are unpredictable, unclonable, and tamper evident ... we will evaluate the capability of the proposed RPPUF to resist ML attacks.” (see Page 14, new Section 4.6, Paragraph 1)
-------Description for new Figure 13.
“To evaluate the ML-resistance of the proposed RPPUF architecture, we utilize ... using these ML attack models and the prediction rates are shown in Fig. 13.” (see Page 14, new Section 4.6, Paragraph 2)
“In Fig. 13, it is obviously that RPPUF exhibits a little better resistance to ... thereby enabling them to achieve high accuracy. .” (see Page 14, new Section 4.6, Paragraph 3)
-------Description for new Table 5 and 6.
“The final prediction rates, including ... the attack success efficiency for DT is the highest.” (see Page 14 - 15, new Section 4.6, Paragraph 4)
“In general, the ML-resistance of the proposed RPPUF is ... it needs to be further strengthened.” (see Page 15, new Section 4.6, Paragraph 2)

Round 3
Reviewer 1 Report
Thanks for the authors for making another round of editing on the manuscript. The added machine learning resistance experimental results part looks good. The prediction results provide some evidence that the proposed PUF is ML resistant. However, I have some suggestions on the results.
1) It looks that Figure 13 is a comparison between RPPUF and arbiter PUF, right? Could the authors make it clear in the annotation of the figure?
2) The authors claims that "the RPPUF exhibits a little better resistance to ANN attacks". Usually ANN is a more powerful method than other ML methods to attack strong PUFs. So the authors conclusion seems not reasonable to me. Would the authors provide more details of the structure of the ANN (e.g. layer number, node number, etc)? Similarly, we would like the authors to provide details of structure and parameters of other ML attacks.
3) The prediction results show that the prediction rate can still be upto 68.33%, which is much different from a 50% prediction rate. Could the authors provide some explanation of the results and show why 68.33% is an acceptable number?
Author Response
Response to Reviewer 1 Comments
Original Manuscript ID: Electronics-1421770
Original Article Title: “RPPUF: An Ultra-Lightweight Reconfigurable Pico-Physically Unclonable Function for Resource-constrained IoT Devices”
To: Electronics Editor
Re: Response to Reviewer
Dear Editor,
Thank you for allowing a re-submission of our manuscript, with an opportunity to address the reviewers’ comments. We appreciate editor and reviewers very much for their positive and constructive comments and suggestions on our manuscript entitled “RPPUF: An Ultra-Lightweight Reconfigurable Pico-Physically Unclonable Function for Resource-constrained IoT Devices”.
We are uploading (a) our point-by-point response to the comments (below) (response to reviewers), (b) a updated manuscript with modification highlights (PDF main document), and (c) a clean updated manuscript without highlights (PDF main document).
We have studied reviewer’s comments carefully and have made a revision. We have tried our best to revise our manuscript according to the comments. Thank you and best regards.
Best regards,
Zhao Huang et al.
Corresponding Author:
Quan Wang
qwang@xidian.edu.cn
Reviewer:
Thanks for the authors for making another round of editing on the manuscript. The added machine learning resistance experimental results part looks good. The prediction results provide some evidence that the proposed PUF is ML resistant. However, I have some suggestions on the results.
1) It looks that Figure 13 is a comparison between RPPUF and arbiter PUF, right? Could the authors make it clear in the annotation of the figure?
2) The authors claims that "the RPPUF exhibits a little better resistance to ANN attacks". Usually ANN is a more powerful method than other ML methods to attack strong PUFs. So the authors conclusion seems not reasonable to me. Would the authors provide more details of the structure of the ANN (e.g. layer number, node number, etc.)? Similarly, we would like the authors to provide details of structure and parameters of other ML attacks.
3) The prediction results show that the prediction rate can still be up to 68.33%, which is much different from a 50% prediction rate. Could the authors provide some explanation of the results and show why 68.33% is an acceptable number?
To Reviewer:
The authors appreciate very much for your constructive comments on our manuscript, “RPPUF: An Ultra-Lightweight Reconfigurable Pico-Physically Unclonable Function for Resource-constrained IoT Devices” (Manuscript ID Electronics-1421770).
We have revised our manuscript according to your comments. The revision details are summarized as follows.
Point # 1: It looks that Figure 13 is a comparison between RPPUF and arbiter PUF, right? Could the authors make it clear in the annotation of the figure.
Response 1: We are appreciative of the reviewer’s suggestion. Yes, Figure 13 shows the comparison results between RPPUF and APUF. Just like what the reviewer suggested, we separate the original Figure 13 into 5 sub-figure and re-annotate them in detailed..
Action 1: To carefully address your comment, we have revised and updated the manuscript as follows.
- Separate the original Fig. 13 into 5 new sub-figures, Fig. 13(a) - (e). (see Page 14, Section 7, new Fig. 13(a)-(e))
- Re-annotate each sub figure more clearly.
For more details, please kindly refer to our revised manuscript, thank you.
Point # 2: The authors claims that "the RPPUF exhibits a little better resistance to ANN attacks". Usually ANN is a more powerful method than other ML methods to attack strong PUFs. So the authors conclusion seems not reasonable to me. Would the authors provide more details of the structure of the ANN (e.g. layer number, node number, etc.)? Similarly, we would like the authors to provide details of structure and parameters of other ML attacks.
Response 2: We are very sorry for our negligence on of the prediction rate on the ANN attack results for RPPUF. Since the ANN attack model we utilized at the very beginning was too simple, we have the previous conclusions. Thus, we re-perform the ANN attack on RPPUF using the parameter settings in new Fig. 5 and update our manuscript with the correct results.
Action 2: To carefully solve your comment, we have updated the manuscript and conducted the following revisions.
- Update the new Fig. 13(a). (see Page 14, Section 6, new Fig. 13(a) )
- Update the corresponding Tables (i.e., current Table 6-7, 10).
(see Page 15, Section 4.6, current Table 6-7 )
(see Page 17, Section 4.7, current Table 10)
- Add new Figure 5 to show the parameter settings of each ML attack models. (see Page 15, Section 4.6, new Table 5).
- Provide corresponding descriptions about new Table 5 (see Page 14, Section 4.6, Paragraph 2, Line 4-9).
“These five ML models are implemented using Open source Python packages scikit-learn and ... The specific parameter settings of other ML models are listed in Table 5.”
- Update corresponding descriptions of the prediction results about ANN.
“... However, both APUF and RPPUF fail to resist ANN attack...” (see Page 15, Section 4.6, Paragraph 1, last Line 1)
“However, the APR value for each size of training CRP sample sets varies from 80.87% to 86.41%, while the MaxPR value...” (see Page 15, Section 4.6, Paragraph 2, Line 6-7)
“... From Table 7, it is evident that ANN attack consumes the longest training time, although it exhibits good prediction results ...” (see Page 15, Section 4.6, Paragraph 2, Line 8-9)
“... However, the FD-APUF can achieve the APR values of more than 86.8% by using ANN and ES ...” (see Page 17, Section 4.7, Paragraph 2, Line 9-10)
“... It is evident that the APR of the RPPUF can achieve the prediction rate vales within 62.5% - 64.5% by applying SVM and LR. However, the attack results for ANN, DT and RF are over 97.8%, 99% and 99.6%, respectively ...” (see Page 17, Section 4.7, Paragraph 2, last Line 3-6)
Point # 3: The prediction results show that the prediction rate can still be up to 68.33%, which is much different from a 50% prediction rate. Could the authors provide some explanation of the results and show why 68.33% is an acceptable number?
Response 3: We are very sorry for our negligence of inappropriate descriptions on the prediction results of ML attacks on RPPUF. Here we mainly want to express that the RPPUF has a better attack resistance on LR and SVM in various ML attacks, with an average prediction rate of 62.83% and 64.24% respectively. And the prediction results for LR and SVM are closer to the ideal value of 50% when compared to ANN, DT and RF. However, current descriptions may be confusing to readers..
Action 3: To carefully address your comment, we have revised and updated the manuscript as follows.
- Update the new Fig. 13(a). (see Page 14, Section 6, new Fig. 13(a) )
- Revise the corresponding descriptions of the prediction results on ANN attack according to the new Figure 13(a).
“From Fig. 13, it is obviously that RPPUF exhibits a little better resistance to LR attack when compared to...” (see Page 14, Section 4.6, Last Paragraph 1, Line 1-3)
“... Similarly, the prediction rate of SVM attack for RPPUF structure is close to LR, and the maximum value for them is not yet beyond 68%....” (see Page 14, Section 4.6, Last Paragraph 1, Line 3-4)
- Provide some explanation of the results to present the conclusion according to Fig 13(a)-(e).
“... Since the ideal prediction rate of ML attack-resistant strong PUF is 50%, our RPPUF is more resistant to LR and SVM attacks compared to the other three ML models. ....” (see Page 14-15, Section 4.6, Last Paragraph 1, Line 3-4)
- Add the description about the comparison of RPPUF and APUF.
“... In contrast, it is almost powerless for APUF to resist SVM and LR attacks. This result illustrates that APUF and RPPUF have different resistance to LR and SVM attacks. ...” (see Page 15, Section 4.6, Paragraph 1, Line 2-4)
